# Dimension-free Private Mean Estimation for Anisotropic Distributions

**Yuval Dagan**
School of Computer Science
Tel Aviv University
ydagan@tauex.tau.ac.il

**Michael I. Jordan**
Department of EECS and Statistics
University of California Berkeley
michael_jordan@berkeley.edu

**Xuelin Yang**
Department of Statistics
University of California Berkeley
xuelin@berkeley.edu

**Lydia Zakynthinou**
Department of EECS
University of California Berkeley
lydiazak@berkeley.edu

**Nikita Zhivotovskiy**
Department of Statistics
University of California Berkeley
zhivotovskiy@berkeley.edu

## Abstract

We present differentially private algorithms for high-dimensional mean estimation. Previous private estimators on distributions over $\mathbb{R}^d$ suffer from a curse of dimensionality, as they require $\Omega(d^{1/2})$ samples to achieve non-trivial error, even in cases where $O(1)$ samples suffice without privacy. This rate is unavoidable when the distribution is isotropic, namely, when the covariance is a multiple of the identity matrix. Yet, real-world data is often highly anisotropic, with signals concentrated on a small number of principal components. We develop estimators that are appropriate for such signals—our estimators are $(\varepsilon, \delta)$-differentially private and have sample complexity that is dimension-independent for anisotropic subgaussian distributions. Given $n$ samples from a distribution with known covariance-proxy $\Sigma$ and unknown mean $\mu$, we present an estimator $\hat{\mu}$ that achieves error , $\|\hat{\mu} - \mu\|_2 \leq \alpha$, as long as $n \gtrsim \operatorname{tr}(\Sigma)/\alpha^2 + \operatorname{tr}(\Sigma^{1/2})/(\alpha\varepsilon)$. We show that this is the optimal sample complexity for this task up to logarithmic factors. Moreover, for the case of unknown covariance, we present an algorithm whose sample complexity has improved dependence on the dimension, from $d^{1/2}$ to $d^{1/4}$.

## 1 Introduction

Machine learning is increasingly deployed in real-world settings to learn about properties of populations, both large and small. When the data comes from human populations, it is essential that algorithm design allows inferring properties of populations without revealing potentially sensitive information about specific individuals in the population. That sensitive information can be revealed, inadvertently or adversarially, has been demonstrated in numerous ways, including via reconstruction attacks [21, 24], as well as membership-inference attacks [56], often targeting sensitive genomic data [34, 55, 64]. To mitigate the risk of privacy violations in general database theory, Dwork, McSherry, Nissim, and Smith [25] proposed the rigorous guarantee of *differential privacy* (DP), which has

---

*Authors ordered alphabetically.

38th Conference on Neural Information Processing Systems (NeurIPS 2024).

been widely adopted in industry [29, 11, 33, 62, 53, 4] and government [30, 1, 2]. Algorithms that are differentially private are guaranteed to not leak too much information about the individuals in a database.

In the machine learning setting, there is a tension between differential privacy and inferential and predictive accuracy. It is an ongoing challenge to capture that tension mathematically, in a way that is applicable to a wide variety of problems and is sufficiently quantitative so as to provide a guide for real users and real systems designers. A particularly salient theoretical challenge is to obtain results that capture dimension-dependence—given that machine learning data are often of high dimensionality and involve significant correlation among dimensions, and given that privacy is difficult to guarantee in high dimensions, particularly so when there are correlations. Indeed, differentially private inference suffers from a *curse of dimensionality*—the sample size $n$ that is required to obtain a non-trivial DP learner is often polynomial in the dimension $d$ of the data.

Significant progress has been made in addressing this challenge in recent years by focusing on a relatively simple inferential task, that of high-dimensional mean estimation. Formally, given a data set of $n$ points, $X = (X^{(1)}, \ldots, X^{(n)}) \in \mathbb{R}^{d \times n}$ drawn i.i.d. from a multivariate distribution $\mathcal{P}$ with unknown mean $\mu \in \mathbb{R}^d$, the goal is to learn $\mu$.

Obtaining low-error private mean estimators in the high-dimensional regime is not always possible. For example, consider a Gaussian distribution $\mathcal{P} = \mathcal{N}(\mu, \sigma^2 I_d)$, where $I_d$ is the $d \times d$ identity matrix. Here, the sample complexity of any private estimator $\hat{\mu}$ achieving error $\|\hat{\mu} - \mu\|_2 \leq \alpha$ is $n = \Omega(d\sigma^2/\alpha^2 + d\sigma/(\alpha\varepsilon))$ [39], where $\varepsilon$ is the privacy parameter.[1] The first term corresponds to the non-private sample complexity and the second term to the additional samples required due to privacy. Although both depend on $d$, note that for non-trivial error $\alpha = 0.01\sigma\sqrt{d}$ and $\varepsilon = 0.1$, the non-private term is $O(1)$, whereas the dimension-dependence persists in the cost of privacy which is $O(\sqrt{d})$.

In spite of this lower bound, there is still hope for obtaining better dependence on the dimension in certain cases. This is due to the fact that the lower bound instance assumes that the covariance is isotropic: a multiple of the identity matrix. However, real-world data are far from being isotropic. Often, the signal is concentrated in a few directions, while it is significantly weaker in others, as can be revealed via Singular Value Decomposition (SVD). In these cases, there are several examples of non-private estimators for a variety of tasks which exploit the structure of the data to achieve lower sample complexity. Specifically for mean estimation of Gaussian distributions, as in our example above, only $n = O(\text{tr}(\Sigma)/\alpha^2)$ samples are required [48] (this number of samples is sufficient even for robust estimators under the strong contamination model [50]). This bound is instance-adaptive, as the trace of the covariance matrix $\text{tr}(\Sigma)$ equals its upper bound, $d\|\Sigma\|_2$, in the isotropic case, but can be much smaller for anisotropic data. Exploiting the non-isotropic structure of the covariance matrix is also central to the covariance estimation problem with respect to the operator norm (namely, when the error between the true covariance matrix $\Sigma$ and its estimate $\hat{\Sigma}$ is measured in terms of $\|\hat{\Sigma} - \Sigma\|_2$) [43, 65]. A more recent focus is on overparametrized linear regression [9], where again the highly non-isotropic structure of the covariance matrix allows for inference under certain assumptions on the decay of eigenvalues of the covariance matrix. In all the mentioned results, non-private estimation is possible when $n \ll d$, including even infinite-dimensional Hilbert spaces.

Returning to private estimation, prior work has obtained optimal bounds for learning the mean of high-dimensional (sub)Gaussian distributions in the affine-invariant Mahalanobis distance [18, 39, 3, 46, 13, 47]. These imply an upper bound for learning the mean in Euclidean distance in the order of $n = O(d\|\Sigma\|_2/\alpha^2 + d\sqrt{\|\Sigma\|_2}/(\alpha\varepsilon))$, which is optimal for isotropic, but loose for anisotropic cases. A folklore estimator, based on [41] achieves $n = \Omega(\sqrt{d\,\text{tr}(\Sigma)}/(\alpha\varepsilon))$, while [8] are the first to focus on the anisotropic case and obtain improved bounds for diagonal covariance, achieving error $n = \Omega(\text{tr}(\Sigma^{1/2})/(\alpha\varepsilon) + \sqrt{d}/\varepsilon)$. Thus, all previous work requires that the sample complexity is at least $\Omega(\sqrt{d})$, which excludes the high-dimensional scenario we are interested in. We are led to pose the following question.:

---

[1]We focus on *approximate* $(\varepsilon, \delta)$-DP, as opposed to *pure* $(\varepsilon, 0)$-DP. However, we omit any dependence on $\delta$ in the introduction.

**Question 1.** *Is it possible to obtain good private mean estimators with a sample size that grows slower with the dimension, or is even dimension-independent, when the covariance of the data is far from isotropic? What is the optimal sample complexity in the case of known and unknown covariance?*

## 1.1 Our contributions

First, note that no improved bounds are possible for *pure* DP, as follows directly from the so-called *packing* technique [32, 18] and specifically applying [18, Lemma 5.1]: any $\varepsilon$-DP algorithm which estimates the mean of a Gaussian distribution up to constant accuracy requires $n = \Omega\left(d/\varepsilon\right)$ samples. This negative result motivates us to focus on $(\varepsilon, \delta)$-differential privacy.

In order to make progress, one would like to utilize the fact that when the covariance is far from being isotropic, the data is closer to being low-dimensional. Concretely, let $\Sigma$ be the covariance matrix of $\mathcal{P}$ and $\sigma_1^2 \geq \ldots \geq \sigma_d^2$ its singular values. If the covariance is far from isotropic, there are only few directions with non-trivial variance. For illustration, if $\sigma_1 = \cdots = \sigma_k = 1$, whereas $\sigma_{k+1} = \cdots = \sigma_d = 1/d$, then the distribution is, in some sense, close to being $k$-dimensional. Here, we would like our sample complexity to be of order $k$ rather than $\sqrt{d}$.

We start by presenting a result in the case where the covariance matrix is known. Here, the bound depends *only* on $\sum_{i=1}^{d} \sigma_i = \mathrm{tr}(\Sigma^{1/2})$, a quantity allowing less contribution from small singular values:

**Theorem 1.1** (Upper bound, known covariance, informal). *Set $\varepsilon, \delta \in (0, 1)$, $\alpha > 0$. Let $X \sim \mathcal{N}(\mu, \Sigma)^n$ with known covariance. There exists an $(\varepsilon, \delta)$-differentially private algorithm which, with probability 0.99, returns an estimate $\hat{\mu}$ such that $\|\hat{\mu} - \mu\|_2 \leq \alpha$, and has sample complexity*

$$n = \tilde{O}\left(\frac{\mathrm{tr}(\Sigma)}{\alpha^2} + \frac{\mathrm{tr}(\Sigma^{1/2})\sqrt{\log(1/\delta)}}{\alpha\varepsilon} + \frac{\log(1/\delta)}{\varepsilon}\right). \tag{1}$$

The first term corresponds to the non-private sample complexity, whereas the remaining two terms are due to privacy. The result extends to subgaussian distributions. In the example illustrated above, this bound indeed yields a dimension-independent complexity of $n = \tilde{O}_\delta(k/\alpha^2 + k/(\alpha\varepsilon))$.

We show that the sample complexity of Theorem 1.1 is nearly optimal. Indeed, the first summand is optimal due to [19, Theorem 4], while the last summand is optimal by a lower bound in the univariate case [41]. We show the optimality of the intermediate summand in (1) up to polylogarithmic terms.

**Theorem 1.2** (Lower bound, informal). *Any $(\varepsilon, \delta)$-DP algorithm which estimates the mean $\mu \in [-1, 1]^d$ of a Gaussian up to $\alpha$ with probability 0.99 has sample complexity $n = \Omega\left(\frac{\mathrm{tr}(\Sigma^{1/2})}{\alpha\varepsilon \log^2(d)}\right)$.*

We now move to the case of unknown covariance. A first approach would be to learn the covariance approximately, namely, find a matrix $A$ such that $A \preceq \Sigma \preceq CA$, for some $C > 1$, and then use $A$ instead of $\Sigma$ in our known-covariance estimator. However, learning such a matrix $A$ privately requires sample size $n = \Theta(d^{3/2})$ [39, 40]. Another approach would be to learn only the diagonal elements of the covariance [41] this would require $n = O(\sqrt{d}/\varepsilon)$ samples. Below, we obtain a sample complexity whose dependence in the dimension is $d^{1/4}$, together with a dependendence on the diagonal elements of the covariance matrix:

**Theorem 1.3** (Upper bound, unknown covariance, informal). *Let parameters $\varepsilon, \delta \in (0, 1)$. Let $X \sim \mathcal{N}(\mu, \Sigma)^n$ with unknown covariance $\Sigma$. There exists an $(\varepsilon, \delta)$-DP algorithm which, with probability 0.99, returns an estimate $\hat{\mu}$ such that $\|\hat{\mu} - \mu\|_2 \leq \alpha$, and has sample complexity*

$$n = \tilde{O}\left(\frac{\mathrm{tr}(\Sigma)}{\alpha^2} + \frac{\sum_{i=1}^{d} \Sigma_{ii}^{1/2}\sqrt{\log(1/\delta)}}{\alpha\varepsilon} + \frac{d^{1/4}\sqrt{\sum_{i=1}^{d} \Sigma_{ii}^{1/2}}\log(1/\delta)}{\sqrt{\alpha}\varepsilon}\right). \tag{2}$$

In general, $\mathrm{tr}(\Sigma^{1/2}) \leq \sum_{i=1}^{d} \Sigma_{ii}^{1/2}$, and if $\Sigma$ is diagonal, the two quantities coincide. Our theorem is in fact more adaptable to easier cases of covariance structure. As a special case, when the covariance is diagonal and the singular values exhibit an exponential decay, that is, $\sigma_i = \sigma_1 e^{-(i-1)}$, then $n = \tilde{O}\left(\frac{\mathrm{tr}(\Sigma)}{\alpha^2} + \frac{\mathrm{tr}(\Sigma^{1/2})\sqrt{\log(1/\delta)}}{\alpha\varepsilon} + \frac{\log^{5/3}(d)\log^{3/2}(1/\delta)}{\varepsilon}\right)$ samples suffice even under unknown covariance.

## 1.2  Techniques

**Known covariance.**  A folklore $(\varepsilon, \delta)$-DP algorithm, based on techniques for the univariate case developed by [41], is to filter outliers by privately estimating each individual coordinate of the mean, $\mu_i$, up to an additive error of $\tilde{O}(\Sigma_{ii}^{1/2})$ for all $i$, clipping any sample point to within that range, and outputting the mean of the modified data set with added spherical Gaussian noise $\mathcal{N}(0, \mathrm{tr}(\Sigma)I_d/(\varepsilon^2 n^2))$.

A standard analysis of this procedure yields a sample complexity of $n \gtrsim \frac{\mathrm{tr}(\Sigma)}{\alpha^2} + \frac{\sqrt{d}}{\varepsilon} + \frac{\sqrt{d\,\mathrm{tr}(\Sigma)}}{\alpha\varepsilon}$, where the dependence on $\delta$ is omitted for clarity. For constant $\varepsilon$, the folklore estimator achieves *privacy for free*, that is, the error due to privacy is lower than the error of statistical estimation, when $n \gtrsim d$.

An improvement to this simple analysis, proposed recently by Aumüller et al. [8] for matrices of diagonal covariance, suggests adding noise $\mathcal{N}\big(0, \mathrm{tr}(\Sigma^{1/2})\Sigma^{1/2}/(\varepsilon^2 n^2)\big)$ instead, which introduces more noise in the directions of larger variance. Slightly simplifying their result and additionally ignoring logarithmic factors in $d$, and the range of $\mu$, their sample complexity is $n \gtrsim \frac{\mathrm{tr}(\Sigma)}{\alpha^2} + \frac{\sqrt{d}}{\varepsilon} + \frac{\mathrm{tr}(\Sigma^{1/2})}{\alpha\varepsilon}$. This estimator achieves privacy for free as long as $n \gtrsim \max\{\|\boldsymbol{\sigma}\|_1^2/\|\boldsymbol{\sigma}\|_2^2, \sqrt{d}\}$, where $\boldsymbol{\sigma}^2$ denotes the vector of singular values of $\Sigma$. While this removes the dimension dependence in the third term compared to the naïve sample complexity, the second term still requires $\Omega(\sqrt{d})$ samples. This is due to the first step of the algorithm (inherited from [37]), which performs $d$ independent estimation tasks. In both approaches, the pre-processing step is a form of coarse mean estimation which ensures that the data will not include outliers, and it is the source of sample-inefficiency.

Thus, in our work, we remove outliers, namely vectors too far away from the true mean in one of the coordinates, using only $n = \tilde{O}(1/\varepsilon)$ samples, thus completely removing the dependence on $d$ in the final sample complexity bounds. (Indeed, our estimator achieves privacy for free for $n \gtrsim \|\boldsymbol{\sigma}\|_1^2/\|\boldsymbol{\sigma}\|_2^2$.) Next, we generalize the approach of [8] to general covariance, rather than diagonal. Finally, we show that the sample complexity is nearly optimal. Specifically:

Our pre-processing is realized by using a polynomial-time filtering algorithm of Tsfadia et al. [63]. Given a predicate computed for two data points, so-called $\mathrm{FriendlyCore}$ returns a subset $X'$ of the input, such that all pairs of the remaining, unfiltered data points satisfy the predicate. Its sample complexity is $\tilde{O}(1/\varepsilon)$ for *any* predicate, hence it has the potential to yield a dimension-independent bound. For our purposes, $X'$ needs to satisfy some sensitivity properties. It follows from our analysis that the filtering should be such that for any two points $X^{(j)}, X^{(\ell)} \in X'$, $\|\Sigma^{-1/4}(X^{(j)} - X^{(\ell)})\|_2^2 \leq \tilde{O}\big(\mathrm{tr}(\Sigma^{1/2})\big)$.

The lower bound for $(\varepsilon, \delta)$-DP is an application of the standard fingerprinting [15, 28, 39, 40] technique for isotropic Gaussians. A straightforward modification of the technique to anisotropic covariance $\Sigma$ gives a weaker bound than Theorem 1.2. Instead one needs to choose an appropriate set of almost-isotropic coordinates whose size scales with $\mathrm{tr}(\Sigma^{1/2})$, and apply the technique to that set.

**Unknown covariance.**  Moving to the case of unknown covariance, for illustration, we focus on the simpler, yet fundamental, case where the covariance matrix is diagonal, so that $\Sigma = \mathrm{diag}(\sigma_1^2, \ldots, \sigma_d^2)$. First, the folklore algorithm described in the known-covariance setting, which adds spherical Gaussian noise, does not require knowledge of the covariance but only of its trace. The trace can be privately learned with $n = \tilde{O}(1/\varepsilon)$ samples. Second, we note that with $n = \tilde{O}(\sqrt{d}/\varepsilon)$ samples, it is possible to learn each $\sigma_i$ up to a multiplicative constant [41]. This allows us to apply the algorithm with known covariance from Theorem 1.1. However, the first step in this approach still requires $\Omega(\sqrt{d})$ samples.

Our approach is to combine these two methods. We privately learn the largest $k \approx \varepsilon^2 n^2$ variances, and their indices. This is done using the sparse vector technique [23] and can be achieved with $n$ samples. We use the *known-covariance* algorithm to estimate the mean in these top $k$ coordinates, with the same error bound as in the known-covariance setting. For the mean at the remaining coordinates, we use the algorithm that only requires knowledge of the trace of the covariance. The error of the latter estimate is $\alpha_{\mathrm{bot}} \approx \sqrt{d}\|\boldsymbol{\sigma}_{\mathrm{bot}}\|_2/(n\varepsilon)$, where $\boldsymbol{\sigma}_{\mathrm{bot}}$ is the vector containing the lowest $d - k$ variances. The first observation is that $\boldsymbol{\sigma}$ contains at least $k$ entries as large as $\|\boldsymbol{\sigma}_{\mathrm{bot}}\|_\infty$, hence, $\|\boldsymbol{\sigma}\|_1 \geq k\|\boldsymbol{\sigma}_{\mathrm{bot}}\|_\infty$. Then, by Hölder's inequality, $\|\boldsymbol{\sigma}_{\mathrm{bot}}\|_2 \leq \sqrt{\|\boldsymbol{\sigma}_{\mathrm{bot}}\|_1\|\boldsymbol{\sigma}_{\mathrm{bot}}\|_\infty}$. Substituting $k$ yields $\alpha_{\mathrm{bot}} \approx \sqrt{d}\|\boldsymbol{\sigma}\|_1/(\varepsilon^2 n^2)$, which implies the desired sample complexity bound in Theorem 1.3.

## 1.3 Related work

**Differentially private Gaussian mean estimation.** Smith [58] proposed estimators for asymptotically normal statistics with optimal convergence rates under a certain range of parameters. The optimal sample complexity for learning the mean of a Gaussian with known covariance in *Mahalanobis norm* under $(\varepsilon, \delta)$-DP is $n \gtrsim d/\alpha^2 + d/(\alpha\varepsilon) + \log(1/\delta)/\varepsilon$ and has been established in a series of works [18, 39, 3, 46], starting from [41] in the univariate setting. Given the covariance matrix $\Sigma$, the Mahalanobis distance between the estimate $\tilde{\mu}$ and the true mean $\mu$ is defined as: $\|\tilde{\mu} - \mu\|_\Sigma = \|\Sigma^{-1/2}(\tilde{\mu} - \mu)\|_2$. When $\Sigma = I_d$, the Mahalanobis and Euclidean norms coincide. The Mahalanobis distance yields an affine-invariant accuracy guarantee, and $\|\tilde{\mu} - \mu\|_\Sigma \leq \alpha$ immediately implies $\|\tilde{\mu} - \mu\|_2 \leq \alpha\sqrt{\|\Sigma\|_2}$. However, the power of the Mahalanobis guarantee is overshadowed by the fact that even for $\alpha = \sqrt{d}$, a large sample size, namely $n = \Omega(\sqrt{d})$, is required, which excludes the high-dimensional scenario we are interested in.[2] Furthermore, confidence sets induced by guarantees in the Euclidean distance have the pleasant property of being more easily constructible.

**Beyond global sensitivity.** There are several lines of work within differential privacy which aim to satisfy some form of instance-adaptive accuracy guarantee, as we do. General purpose frameworks which aim to privately estimate a statistic of the data, with error which adapts to "good" data sets, include propose-test-release [22], smooth-sensitivity [52], and Lipschitz extensions [12, 42]. Our method follows the same high-level structure as propose-test-release. The latter has been combined with robust estimators to yield optimal private learners for several tasks [13, 47]. Even more generally, [6, 35] give a black-box method which transforms robust estimators to private ones via the *inverse-sensitivity* mechanism [5] (see [59] for a discussion on inverse-sensitivity). As there exist optimal robust estimators for the mean of anisotropic Gaussians [50], this would be a viable approach, but the volumetric analysis of the transformation involves terms which depend on the dimension. Tsfadia et al. [63] propose a filtering method which yields private aggregators whose error adapts to the *diameter* of the input data set. It is their method that we utilize for our upper bounds. A series of works formalize instance-optimality for private estimation of empirical [5, 37, 20] or population [49, 7] quantities. These are all generally well-suited to our setting but either do not adapt to high dimensions, or a direct application would require $n \gtrsim \sqrt{d\,\mathrm{tr}(\Sigma)}/(\alpha\varepsilon)$.

Nikolov and Tang [51] study instance-optimality specifically for Gaussian noise mechanisms, albeit for data that belong in a bounded convex set. Although this is not the case for Gaussian data, it is worth noting that our error rates match those of [51], which hold for arbitrary distributions over $K$, when the bounded set is $K = \mu + \Sigma^{1/2}\mathcal{B}^d(1)$. Privately learning $K$ however would require more samples.

**Privately learning nuisance parameters.** Karwa and Vadhan [41] learn (a constant multiple of) the variance of a univariate Gaussian using $n = \tilde{O}\left(\log(1/\delta)/\varepsilon\right)$ samples. In high dimensions, privately learning the covariance matrix of a Gaussian in spectral norm requires $n \gtrsim d^{3/2}$ samples [39, 40], which is more than one needs to learn the mean under known covariance. Brown et al. [13] avoid the bottleneck of private covariance estimation, showing that the sample complexity of Gaussian mean estimation under known covariance with respect to Mahalanobis distance can in fact be matched, even when the covariance is unknown. Their tools also follow the propose-test-release approach and could be modified to fit our setting, but the privacy analysis would still require $n \gtrsim d$. Singhal and Steinke [57] learn a subspace in which the majority of the data lie, which could be used as a pre-processing step, followed by projection. However, to recover the set of top $k$ eigenvectors, they require that there exists a large gap between the two consecutive variances, that is, $\sigma_k \geq \mathrm{poly}(d)\sigma_{k+1}$.

**Comparison with [8].** The paper by Aumüller et al. [8] is the closest work to ours, aiming to find sample-efficient mean estimators with respect to the Euclidean norm in the anisotropic case. Their work focuses on the less general case of diagonal, (almost) known covariance. The sample complexity of their estimator requires $n \gtrsim \sqrt{d}$, whereas our estimator for the known covariance case is dimension-independent, and, as we prove, optimal. However, the focus in [8] is on estimators that satisfy the stricter privacy guarantee of $\rho$-zCDP, which forces the need for dimension-dependent sample size. This is the key contrast with our dimension-free philosophy. As an interesting distinction,

---

[2]This limitation is due to the fact that the Mahalanobis distance equalizes the variance across all directions and forces us to make inferences even in directions where the distribution has particularly small variance.

Aumüller et al. [8] provide accuracy guarantees with respect to the $\ell_p$ norm (the upper bounds) for slightly more general classes of so-called well-concentrated distributions, which include subgaussians. It would be interesting to establish optimal private mean estimation bounds with respect to general $\ell_p$ norms. In fact, the optimal non-private sample complexity of Gaussian mean estimation, with matching upper and lower bounds, with respect to general norms has been established only recently, and it depends on the Gaussian mean width of the set induced by the unit dual ball of the norm [19].

## 2 Preliminaries

We write $[n] = \{1, \ldots, n\}$, $\log$ denotes the natural logarithm, and $\mathcal{B}^d(c, r)$ denotes the $d$-dimensional Euclidean ball with radius $r$ and center $c$. We omit $c$ if $c = 0$.

We introduce *differential privacy* here. We say that $X, X'$ are *neighboring data sets* if either $\exists j \in [|X|]$ such that $X' = X \setminus X^{(j)}$ or $\exists j \in [|X'|]$ such that $X = X' \setminus X'^{(j)}$.[3] Differentially private algorithms have *indistinguishable* output distributions on neighboring data sets.

**Definition 2.1** (($\varepsilon, \delta$)-indistinguishability). *Two distributions $P, Q$ over domain $\mathcal{W}$ are ($\varepsilon, \delta$)-indistinguishable, denoted by $P \approx_{\varepsilon,\delta} Q$, if for any measurable subset $W \subseteq \mathcal{W}$,*

$$\Pr_{w \sim P}[w \in W] \le e^\varepsilon \Pr_{w \sim Q}[w \in W] + \delta \quad \text{and} \quad \Pr_{w \sim Q}[w \in W] \le e^\varepsilon \Pr_{w \sim P}[w \in W] + \delta.$$

**Definition 2.2** (Differential Privacy [25]). *A randomized algorithm $\mathcal{A} \colon \mathcal{X}^* \to \mathcal{W}$ is ($\varepsilon, \delta$)-differentially private if for all neighboring data sets $X, X'$ we have $\mathcal{A}(X) \approx_{\varepsilon,\delta} \mathcal{A}(X')$. We say that algorithm $\mathcal{A}$ satisfies* pure *differential privacy if it satisfies the definition for $\delta = 0$.*

Differential privacy satisfies several useful properties, such as post-processing and composition [25, 27]. For further details and guarantees of standard DP mechanisms, see Section A.

Our estimators will use the $\mathrm{BasicFilter}$ procedure of Tsfadia et al. [63], whose detailed definition is presented in Section 3. They provide a framework which allows us to extend an algorithm which is private *only for "easy" pairs of data sets*, to an algorithm that is private for any worst-case pair. "Easy" pairs of data sets are modelled with respect to a predicate $f$ between two data points:

**Definition 2.3** ($f$-friendly, Def. 1.1 [63]). *Let $X$ be a data set over $\mathcal{X}$ and let $f : \mathcal{X}^2 \to \{0, 1\}$ be a predicate. We say $X$ is $f$-friendly if for all $x, y \in X$ there exists $z \in \mathcal{X}$ such that $f(x, z) = f(z, y) = 1$.*

**Definition 2.4** ($f$-friendly DP, Def. 1.3 [63]). *An algorithm $\mathcal{A}$ is called $f$-friendly ($\varepsilon, \delta$)-DP if for any neighboring data sets $X, X'$, such that $X \cup X'$ is $f$-friendly, $\mathcal{A}(X) \approx_{\varepsilon,\delta} \mathcal{A}(X')$.*

**Theorem 2.5** (Theorem 4.11 [63]). *Let $\mathcal{A}$ be an $f$-friendly ($\varepsilon, \delta$)-DP algorithm. Given data set $X$, let $\boldsymbol{v} = \mathrm{BasicFilter}(X, f, \alpha = 0)$ and $C(X) = \{X^{(j)}\}_{\{j: v_j = 1\}}$. Then $\mathcal{B}(X) := \mathcal{A}(C(X))$ is $(2(e^\varepsilon - 1)\varepsilon, 2e^{\varepsilon + 2(e^\varepsilon - 1)}\delta)$-DP.*

We assume data are drawn from subgaussian distributions, which include Gaussians.

**Definition 2.6** (Subgaussian distributions). *The random vector $X$ with mean $\mu$ is subgaussian with a p.s.d. covariance matrix proxy $\Sigma$ if for any $\lambda$ and any $v \in \mathbb{R}^d$, $\mathbb{E}\, e^{\lambda \langle X - \mu, v \rangle} \le e^{\lambda^2 v^\top \Sigma v / 2}$.*

**Lemma 2.7** (Norm of the subgaussian vector [36, 65]). *Let $X = (X^{(1)}, \ldots, X^{(n)})$ be drawn i.i.d. from the subgaussian distribution with mean $\mu$ and covariance-proxy $\Sigma$. With probability at least $1 - \beta$, $\|\frac{1}{n} \sum_{j=1}^n X^{(j)} - \mu\|_2 \le \sqrt{\mathrm{tr}(\Sigma)/n} + \sqrt{2\|\Sigma\|_2 \log(1/\beta)/n}$.*

## 3 Nearly-matching upper and lower bounds under known covariance

Algorithm 1 proceeds in two simple steps. The first step filters out outliers so that all remaining pairs of data points satisfy the re-scaled distance predicate $\mathrm{dist}_{M,\lambda}$ and, assuming enough data points remain, the second step releases their empirical mean along with appropriate Gaussian noise.

We retrieve the folklore result, by taking $M = I_d, \lambda \approx \sqrt{\mathrm{tr}(\Sigma)}$, which is known (otherwise, can be easily privately estimated as in Section 4). The filtering then guarantees that all pairs of points

---

[3]This is the so-called add/remove model of DP, which will be convenient for our use of prior work. The same privacy guarantees will also hold for the swap model, where $\mathrm{d}_{\mathrm{Ham}}(X, X') \le 1$, up to constant factors.

are within distance $\sqrt{\operatorname{tr}(\Sigma)}$, and adds spherical Gaussian noise with covariance $\operatorname{tr}(\Sigma)I_d/(\varepsilon^2 n^2)$. To retrieve the optimal bound, take $M = \Sigma$, which splits the privacy budget unevenly among coordinates. Then, $\lambda \approx \sqrt{\operatorname{tr}(\Sigma^{1/2})}$ and the Gaussian noise has covariance $\operatorname{tr}(\Sigma^{1/2})\Sigma^{1/2}/(\varepsilon^2 n^2)$, as in [8].

---

**Algorithm 1** Private Re-scaled Averaging: $\operatorname{Avg}_{M,\lambda,\varepsilon,\delta}(X)$

---

**Require:** Data set $X = (X^{(1)}, \ldots, X^{(n)})^T \in \mathbb{R}^{n \times d}$. Privacy parameters: $\varepsilon, \delta > 0$. Failure probability $\beta > 0$. Symmetric invertible matrix $M$. Parameter $\lambda$.
1: Let $\operatorname{dist}_{M,\lambda}(x, y) = \mathbb{1}\{\|M^{-1/4}(x - y)\|_2 \leq \lambda\}$.
2: $\boldsymbol{v} = \operatorname{BasicFilter}(X, \operatorname{dist}_{M,\lambda}, \alpha = 0)$.
3: Let $C = \{X^{(j)}\}_{\{j:\, v_j=1\}}$.
4: Compute $\hat{n}_C = |C| - \frac{\log(1/\delta)}{\varepsilon} + z$ where $z \sim \operatorname{Lap}(\frac{1}{\varepsilon})$.
5: **if** $|C| = 0$ or $\hat{n}_C \leq 0$ **then**
6:      **return** $\perp$.
7: **return** $\hat{\mu} = \frac{1}{|C|} \sum_{x \in C} x + \eta$ where $\eta \sim \mathcal{N}\left(0, \frac{8 \log(1.25/\delta)\lambda^2}{\varepsilon^2 \hat{n}_C^2} M^{1/2}\right)$.

8: **procedure** $\operatorname{BasicFilter}(X, f, \alpha)$                $\triangleright$ Algorithm 4.3 from [63].
9:      **for** $j = 1, \ldots, n$ **do**
10:          Let $z_j = \sum_{k=1}^n f(X^{(j)}, X^{(k)}) - n/2$.
11:          Sample $v_j = \operatorname{Bern}(p_j)$, where $p_j = \begin{cases} 0, & \text{if } z_j \leq 0, \\ 1, & \text{if } z_j \geq (1/2 - \alpha)n, \\ \frac{z_j}{(1/2-\alpha)n}, & \text{otherwise.} \end{cases}$
12:      **return** $\boldsymbol{v} = (v_1, \ldots, v_n)$

---

**Theorem 3.1.** *Let $\varepsilon \in (0, 10), \delta \in (0, 1), \alpha > 0, \beta \in (0, 1)$. Algorithm 1 is $(\varepsilon, \delta)$-differentially private. Let $X$ be a data set of size $n$, drawn from a subgaussian distribution with covariance-proxy $\Sigma$ and mean $\mu$. Given $M = \Sigma$, $\lambda \geq \sqrt{2\operatorname{tr}(M^{-1/4}\Sigma M^{-1/4})} + 2\sqrt{2\|M^{-1/4}\Sigma M^{-1/4}\|_2 \log(\frac{n}{\beta})}$, with probability at least $1 - \beta$, Algorithm 1 returns $\hat{\mu}$ such that $\|\hat{\mu} - \mu\|_2 \leq \alpha$, as long as*

$$n = \tilde{\Omega}\left(\frac{\operatorname{tr}(\Sigma) + \|\Sigma\|_2 \log \frac{1}{\beta}}{\alpha^2} + \frac{\operatorname{tr}(\Sigma^{1/2})\sqrt{\log \frac{1}{\delta}}}{\alpha\varepsilon} + \frac{\sqrt{\|\Sigma\|_2 \log \frac{1}{\delta}} \log \frac{1}{\beta}}{\alpha\varepsilon} + \frac{\log \frac{1}{\delta\beta}}{\varepsilon}\right), \quad (3)$$

*where $\tilde{\Omega}$ hides constants and a log factor of the third term multiplied with itself.*

The theorem holds more generally for any symmetric invertible $M$, and $\lambda$ satisfying the assumptions. We sketch the proof of Theorem 3.1 next. All remaining details are in Appendix B.

*Proof sketch.* We start with the accuracy analysis. First we show that the original dataset $X$ passes through BasicFilter (i.e., $C = X$) with high probability. It suffices to show that each pair $j \neq k \in [n]$, satisfies $\operatorname{dist}_{M,\lambda}(X^{(j)}, X^{(k)}) = 1$ with probability $1 - \beta/n^2$. Observe that for $j \neq k$, $M^{-1/4}(X^{(j)} - X^{(k)})$ is subgaussian with mean 0 and covariance proxy $2M^{-1/4}\Sigma M^{-1/4}$. By Lemma 2.7 and our setting of $\lambda$, indeed $\|M^{-1/4}(X^{(j)} - X^{(k)})\|_2 \leq \lambda$ for each pair with probability $1 - \beta/n^2$. We condition on $C = X$. With high probability by the CDF of the Laplace distribution and since $|C| = n = \Omega(\log(1/\delta\beta)/\varepsilon)$, $\hat{n}_C = \Omega(n)$. Thus, the algorithm does not abort, and returns estimate $\hat{\mu}$. It remains to upper bound the total error of $\hat{\mu}$. This is at most the error of the empirical mean plus the error due to noise $\|\eta\|_2$. By Lemma 2.7, with high probability, the former is $\tilde{O}(\sqrt{\operatorname{tr}(\Sigma)/n})$ and the latter $\tilde{O}(\lambda\sqrt{\operatorname{tr}(M^{1/2})}/(\varepsilon n))$. Substituting $M = \Sigma$, and the value for $\lambda$, the total error becomes $\tilde{O}(\sqrt{\operatorname{tr}(\Sigma)/n} + \operatorname{tr}(\Sigma^{1/2})/(\varepsilon n))$, which yields the stated sample complexity.

The privacy analysis follows the steps of [63, Claim 3.4]. By Theorem 2.5, it suffices to show that lines 4-7 of Algorithm 1, namely, $\mathcal{A}$, are $\operatorname{dist}_{\Sigma,\beta}$-friendly DP. Consider neighboring inputs $X, X'$, differing in the $n$-th data point w.l.o.g., that is $X' = X \setminus X^{(n)}$. Since $\|X| - |X'\| = 1$, by the guarantees of the Laplace mechanism, the r.v.s $\hat{n}_X, \hat{n}_{X'}$ in Line 4 are $(\varepsilon, 0)$-indistinguishable. Moreover, they both satisfy $\hat{n}_X, \hat{n}'_X < |X|$ with probability $1 - \delta/2 > 1/2$. Conditioning on this event for the remainder of the sketch, we can fix $\hat{n}_X = \hat{n}'_X = \hat{n} < |X|$, for some value $\hat{n}$. If $\hat{n} \leq 0$, both runs abort. Otherwise,

it suffices to show that Line 7 adds sufficient noise to maintain privacy. By post-processing, since $M$ is not data-dependent, this is equivalent to ensuring that $\mathcal{N}(M^{-1/4}\frac{1}{|X|}\sum_{i=1}^{|X|}X^{(i)}, v^2 I_d) \approx_{\varepsilon,\delta}$ $\mathcal{N}(M^{-1/4}\frac{1}{|X|-1}\sum_{i=1}^{|X|-1}X^{(i)}, v^2 I_d)$ where $v = (2\lambda/\hat{n})(\sqrt{2\log(1.25/\delta)}/\varepsilon)$. This is true by the guarantees of the Gaussian mechanism applied to $f(X) = M^{-1/4}\sum_{i=1}^{|X|}X^{(i)}/|X|$, whose $\ell_2$-sensitivity for $\text{dist}_{\Sigma,\lambda}$-friendly $X, X'$ can be upper bounded by $2\lambda/|X| \leq 2\lambda/\hat{n}$ (since $0 < \hat{n} < |X|$, by assumption). By composition, $\mathcal{A}$ indeed satisfies $\text{dist}_{M,\lambda}$-friendly $(O(\varepsilon), O(\delta))$-DP. $\qquad\square$

We show that the sample complexity of Theorem 3.1 is optimal. We briefly explain our lower bound construction here. All remaining details are in Appendix C.

*Proof Sketch of Theorem 1.2.* Let $\Sigma = \text{diag}(\boldsymbol{\sigma}^2)$. Assume w.l.o.g. that $\sigma_1^2 \geq \ldots \geq \sigma_d^2$. Partition the set of coordinates into buckets $S_k = \{i \in [d] : \sigma_i \in \sigma_1 \cdot (2^{-k}, 2^{-k+1}]\}, \forall k \in [\log(d)]$ and $S_{\log(d)+1} = [d] \setminus \bigcup_{k\in[\log(d)]} S_k$. We have that $\sum_{k\in[\log(d)+1]}\sum_{i\in S_k}\sigma_i = \|\boldsymbol{\sigma}\|_1$. Consider the bucket $S$ which contributes the most to this sum and let $\sigma_S$ be the maximum variance in this bucket. It must be that $|S| \geq \frac{\|\boldsymbol{\sigma}\|_1}{(\log(d)+1)\sigma_S}$. The lower bound of [39, Theorem 6.5] for isotropic Gaussians, implies that any $(\varepsilon, \delta)$-private mean estimator which returns, with constant probability, an estimate $\hat{\mu}_S$ with error $\alpha$ for the coordinates in $S$ (note that they are all within a factor of 2), requires $n = \Omega(|S|\sigma_S/(\alpha\varepsilon\log(d))) = \Omega(\|\boldsymbol{\sigma}\|_1/(\alpha\varepsilon\log^2(d)))$ samples. As an estimator for the $d$-dimensional Gaussian mean restricted to $S$, would give us such a $\hat{\mu}_S$, the statement follows. $\qquad\square$

## 4 Handling unknown covariance

In this section we consider the case of unknown covariance. First, recall that $\Omega(d^{3/2})$ samples are required to privately learn the covariance matrix in spectral norm [40], which is prohibitive. The lower bound instance is an almost-isotropic Gaussian, which means that anisotropic distributions may circumvent it. Still, the superlinear dependence on $d$ implies that this approach will yield suboptimal sample complexity for mean estimation. Avoiding private covariance estimation, Brown et al. [13] propose a "covariance-aware" private mean estimator which returns the mean with Gaussian noise which scales with the empirical covariance matrix of the data set $\Sigma_X$, as $\mathcal{N}(0, \lambda_M^2 \Sigma_X/(\varepsilon^2 n^2))$ for appropriate factor $\lambda_M^2$. Since adding data-dependent noise can break privacy, a pre-processing step is required to ensure that no outliers exist in the data set with respect to the empirical covariance, roughly ensuring that $\|\Sigma_X^{-1/2}(X^{(k)} - X^{(j)})\|_2 \leq \lambda_M$, for all $j \neq k \in [n]$. In our case, to maintain the accuracy guarantee of the known-covariance case, the Gaussian noise should be $\mathcal{N}(0, \lambda^2 \Sigma_X^{1/2}/(\varepsilon^2 n^2))$ and all data points should satisfy $\|\Sigma_X^{-1/4}(X^{(k)} - X^{(j)})\|_2 \leq \lambda$. Note that $n \geq \text{tr}(\Sigma)/\|\Sigma\|_2$ samples suffice for the empirical covariance to be close to the true covariance $\Sigma$ in spectral norm [43], so applying the algorithm from [13] could maintain accuracy while still allowing a dimension-free sample complexity. Unfortunately, we still cannot use this approach because $n \geq d$ samples are required for the privacy analysis to go though, namely, for neighboring data sets $X, X'$ it holds that $\mathcal{N}(0, \Sigma_X^{1/2}) \approx_{\varepsilon,\delta} \mathcal{N}(0, \Sigma_{X'}^{1/2})$ for $\varepsilon \approx d/n$, which forces us to take $n \geq d/\varepsilon$ samples. The same is true for the follow-up works of [14, 44] which give polynomial-time versions of this algorithm with slightly better statistical guarantees.

Luckily, our accuracy guarantee does not require the variance estimate in all directions to be accurate. For example, consider all directions with variance at most $\|\Sigma\|_2/d$. Adding spherical Gaussian noise to these directions maintains a dimension-free error, without requiring tighter estimates for their variance. Thus, on a high level, our approach for mean estimation in the unknown covariance case is to identify and estimate as many of the top variances as our sample size allows, which turns out to be $k \approx \varepsilon^2 n^2$, while adding spherical Gaussian noise to the remaining ones.

We sketch the proof of the following theorem. All remaining details are in Appendix D.

**Theorem 4.1.** *Let parameters $\varepsilon, \delta \in (0,1)$. Let $X \sim \mathcal{N}(\mu, \Sigma)^n$ with unknown covariance $\Sigma$. Algorithm 2 is $(\varepsilon, \delta)$-differentially private and, with probability $1 - \beta$, returns an estimate $\hat{\mu}$ such*

---

**Algorithm 2** Private Re-scaled Averaging with Unknown Covariance

---

**Require:** Data set $X = (X^{(1)}, \ldots, X^{(2n)})^T \in \mathbb{R}^{2n \times d}$. Privacy parameters: $\varepsilon, \delta > 0$. Failure probability $\beta > 0$.

1: **Require** $n = \Omega\left(\log^2(d) + \log(\frac{1}{\delta\beta})\sqrt{\log(\frac{1}{\delta})}\log(d)/\varepsilon\right)$.

2: Let $k \leftarrow \varepsilon^2 n^2 / \left(\log^2(d)\log(1/\delta)\log^2(1/\delta\beta) + \log(\varepsilon n)\right)$ and $\ell \leftarrow \Theta(\log(d))$.

3: Split the dataset into two equal halves: $X^{\mathrm{var}}$ and $X^{\mathrm{mean}}$.

4: Split $X^{\mathrm{var}}$ into $m = \lfloor \frac{n}{2\ell} \rfloor$ groups of size $2\ell$. Define $X^{(j,r)}$ as the $r$-th sample in the $j$-th group.

5: **for** each group $j = 1$ to $m$ and each dimension $i = 1$ to $d$ **do**

6:     Define $V_i^{(j)} = \frac{1}{2\ell}\sum_{r=1}^{\ell}(X_i^{(j,2r-1)} - X_i^{(j,2r)})^2$.

7: $\hat{R} \leftarrow \mathrm{FindKthLargestVariance}_{\varepsilon,\delta}(V, k)$.

8: $I_{\mathrm{top}} \leftarrow \mathrm{TopVar}_{\varepsilon,\delta}(V, \hat{R}/8, k)$ and $I_{\mathrm{bot}} \leftarrow [d] \setminus I_{\mathrm{top}}$.

9: **for** each $i \in I_{\mathrm{top}}$ **do**

10:     Estimate $\hat{\Sigma}_{ii} \leftarrow \mathrm{VarianceSum}_{\varepsilon',\delta',\beta'}(V, \{i\})$ for $\varepsilon' \leftarrow \frac{\varepsilon}{\sqrt{k\log(1/\delta)}}, \delta' \leftarrow \frac{\delta}{k}, \beta' \leftarrow \frac{\beta}{k}$.

11: Compute $\hat{S}_{\mathrm{bot}} \leftarrow \mathrm{VarianceSum}_{\varepsilon,\delta,\beta}(V, I_{\mathrm{bot}})$.

12: $\hat{\mu}_{\mathrm{top}} \leftarrow \mathrm{Avg}_{M,\lambda,\varepsilon,\delta}(X^{\mathrm{mean}}[I_{\mathrm{top}}])$, where $M = \mathrm{diag}(\{\hat{\Sigma}_{ii}\}_{I_{\mathrm{top}}}), \lambda = \tilde{\Theta}\left(\sqrt{\sum_{i \in I_{\mathrm{top}}} \hat{\Sigma}_{ii}^{1/2}}\right)$.

13: $\hat{\mu}_{\mathrm{bot}} \leftarrow \mathrm{Avg}_{M,\lambda,\varepsilon,\delta}(X^{\mathrm{mean}}[I_{\mathrm{bot}}])$, where $M = I_d, \lambda = \tilde{\Theta}(\sqrt{\hat{S}_{\mathrm{bot}}})$.

14: **return** $(\hat{\mu}_{\mathrm{top}}, \hat{\mu}_{\mathrm{bot}})$

---

*that $\|\hat{\mu} - \mu\|_2 \leq \alpha$, as long as*

$$n = \tilde{\Omega}\left(\log^2(d) + \frac{\mathrm{tr}(\Sigma)}{\alpha^2} + \frac{\sum_{i=1}^{d}\Sigma_{ii}^{1/2}\sqrt{\log\frac{1}{\delta}}}{\alpha\varepsilon} + \frac{d^{1/4}\sqrt{\sum_{i=1}^{d}\Sigma_{ii}^{1/2}}\log^{5/4}(\frac{1}{\delta})\log(d)}{\sqrt{\alpha}\varepsilon}\right), \quad (4)$$

*where the symbol $\tilde{\Omega}$ hides multiplicative logarithmic factors in $1/\beta$.*

Next, we describe Algorithm 2 and introduce some of its subroutines along with their guarantees. All omitted proofs are in Appendix D. Our algorithm receives a data set $X^{(1)}, \ldots, X^{(n)}$, where each $X^{(i)}$ is a $d$-dimensional vector distributed as $\mathcal{N}(\mu, \Sigma)$. The algorithm starts by splitting the dataset into $m = \lfloor n/(2\ell) \rfloor$ groups each of size $2\ell$, where $\ell = \Theta(\log d)$. Denote the elements of each group $j$ by $X^{(j,1)}, \ldots, X^{(j,2\ell)}$. Within each group $j$, for each coordinate $i$, we compute an estimate $V_i^{(j)}$ for $\Sigma_{ii}$: $V_i^{(j)} = \frac{1}{2\ell}\sum_{r=1}^{\ell}(X_i^{(j,2r-1)} - X_i^{(j,2r)})^2$. For convenience, we define what it means for the $V_i^{(j)}$ variables to provide a good estimate of the set of $\{\Sigma_{ii}\}_{i \in [d]}$.

**Definition 4.2.** *Given variances $\Sigma_{11}, \ldots, \Sigma_{dd}$ and given a set of estimates, $V = \{V_i^{(j)}\}_{j \in [m], i \in [d]}$, we say that $V$ is valid if $|\{j : \forall i \in [d], \Sigma_{ii}/2 \leq V_i^{(j)} \leq 2\Sigma_{ii}\}| \geq 4m/5$.*

*Proof sketch of Theorem 4.1.* For ease of notation, we assume that $\Sigma = \mathrm{diag}(\boldsymbol{\sigma}^2)$. We start from the accuracy analysis. Assume $n$ satisfies the sample complexity bound of Eq. (4). By Chernoff bound, since $\ell = \Theta(\log(d))$ and $m = \Omega(\log(1/\beta))$, with probability $1 - \beta$, $V$ is valid. We use the estimates $V_i^{(j)}$ as inputs to private procedures: $\mathrm{FindKthLargestVariance}(V, k)$ (compute the $k$-th largest variance up to a multiplicative constant), $\mathrm{VarianceSum}(V, I)$ (compute the sum $\sum_{i \in I} \sigma_i^2$ up to a multiplicative constant), and $\mathrm{TopVar}(V, R, k)$ (identify the indices of the at-most-$k$ largest variances $\sigma_i^2 \geq R$). The first two tasks can be implemented by the Stable Histogram algorithm [16] if $m = \Omega(\log(1/\delta)/\varepsilon)$. TopVar can be implemented via the Sparse Vector technique [26, 54, 31], if $m = \Omega(\sqrt{k\log(1/\delta)}\log(d/\beta)/\sqrt{\varepsilon n})$. Both are satisfied for the given $m = n/2\ell$. With these procedures, the algorithm privately learns the $k$-th largest variance $R$, identifies the top $k$ coordinates, $I_{\mathrm{top}}$, and learns estimates $\{\hat{\sigma}_i\}_{I_{\mathrm{top}}}$ that are accurate up to a multiplicative constant.

Next, we estimate the mean $\mu$ in the coordinates $I_{\mathrm{top}}$ separately from $I_{\mathrm{bot}} := [d] \setminus I_{\mathrm{top}}$, denoted by $\mu_{\mathrm{top}}$ and $\mu_{\mathrm{bot}}$, respectively. Denote vector $\boldsymbol{\sigma}_{\mathrm{top}} = (\{\sigma_i\}_{i \in I_{\mathrm{top}}})$ and $\boldsymbol{\sigma}_{\mathrm{bot}} = (\{\sigma_i\}_{i \in I_{\mathrm{bot}}})$. To estimate $\mu_{\mathrm{top}}$, Algorithm 1, given input vectors $X^{(1)}, \ldots, X^{(n)}$, restricted to coordinates $I_{\mathrm{top}}$,

diagonal matrix $M$, with $M_{ii} = \hat{\sigma}_i^2$ (assume that the rows and columns of $M$ are indexed by $I_{\text{top}}$), and $\lambda \approx \sqrt{\|\boldsymbol{\sigma}_{\text{top}}\|_1}$, returns an estimate $\hat{\mu}_{\text{top}}$ with error $\alpha$ (since the sample complexity of Eq. (4) is larger than the one required by Theorem 3.1).

To estimate $\mu_{\text{bot}}$, we do not know the variances, so we use the naïve approach. We first call VarianceSum once again, to provide an estimate $\hat{t}$ of $\|\boldsymbol{\sigma}_{\text{bot}}\|_2$ up to a multiplicative constant. Given $\hat{t}$, we again call Algorithm 1, now for a $(d-k)$-dimensional estimation problem. Given input vectors $X^{(1)}, \ldots, X^{(n)}$, restricted to coordinates $I_{\text{bot}}$, matrix $M = I_{d-k}$, and $\lambda \approx \hat{t}$, Algorithm 1 returns an estimate $\hat{\mu}_{\text{bot}}$ with error $\alpha$ as long as $n = \tilde{\Omega}(\sqrt{d}\|\boldsymbol{\sigma}_{\text{bot}}\|_2/(\alpha\varepsilon))$. By Hölder's inequality, $\|\boldsymbol{\sigma}_{\text{bot}}\|_2 \leq \sqrt{\|\boldsymbol{\sigma}_{\text{bot}}\|_1\|\boldsymbol{\sigma}_{\text{bot}}\|_\infty} \leq \sqrt{\|\boldsymbol{\sigma}\|_1\|\boldsymbol{\sigma}_{\text{bot}}\|_\infty}$. By the guarantees of TopVar and FindKthLargestVariance, $\|\boldsymbol{\sigma}_{\text{bot}}\|_\infty$ is smaller than the $k$-th largest variance of $\Sigma$ up to a multiplicative constant, which, in turn, must be smaller than $\|\boldsymbol{\sigma}\|_1/k$. Substituting this above, we obtain that it suffices for the stated sample complexity to additionally satisfy $n = \tilde{\Omega}(\sqrt{d}\|\boldsymbol{\sigma}\|_1/(\sqrt{k}\alpha\varepsilon))$, which can be confirmed by substituting the definition for $k$.

The privacy guarantee follows directly by composition of $O(1)$ $(\varepsilon, \delta)$-DP mechanisms. □

*Remark* 1. We note that the sample complexity of Algorithm 2 in fact depends on the decay of the diagonal elements of $\Sigma$, and can yield improved bounds for easier instances. In particular, the error of the algorithm due to privacy is in the order of $\|\boldsymbol{\sigma}_{I_{\text{top}}}\|_1/(\varepsilon n) + \sqrt{|I_{\text{bot}}|}\|\boldsymbol{\sigma}_{I_{\text{bot}}}\|_2/(\varepsilon n)$. Thus, if $\boldsymbol{\sigma}$ follows an exponential decay, i.e., the $i$-th largest variance is proportional to $e^{-(i-1)}$, or all $\boldsymbol{\sigma}_{\text{bot}}$ variances are smaller than $\|\boldsymbol{\sigma}\|_1/d$, then it suffices to learn only the top $k = \log(d)$ variances and the error almost matches that of the known-covariance case, up to additional logarithmic factors in $d$, $1/\delta$. Moreover, identifying easier instances is possible by computing a private histogram over $\log(d)$ buckets of the form $(2^{-j}, 2^{-j+1}]\|\boldsymbol{\sigma}\|_\infty$, given $n = \tilde{O}(\log(d)/\varepsilon)$ samples [16, 41].

Thus, we can determine special cases where the decay of $\Sigma$ allows us to achieve the optimal rate of Theorem 1.1 even with unknown diagonal covariance. But without further assumptions, our algorithm has sample complexity that depends on $d^{1/4}$. The question of the optimal sample complexity for mean estimation in the case of unknown covariance, which captures anisotropic distributions, remains open.

## 5   Conclusion and future work

We present $(\varepsilon, \delta)$-differentially private mean estimators for subgaussian distributions with error $\alpha$ as measured in Euclidean distance, with high probability, as long as the sample size is $n = \tilde{\Theta}\left(\text{tr}(\Sigma)/\alpha^2 + \text{tr}(\Sigma^{1/2})/(\alpha\varepsilon)\right)$. The sample complexity is thus dimension-independent when the covariance is highly anisotropic. We show that this is the optimal sample complexity for this task up to logarithmic factors. We also present an algorithm in the more challenging case of unknown covariance, whose sample complexity has improved dependence on the dimension, that is, $d^{1/4}$.

In the known covariance case, the dependence on $\log(1/\delta)$ could possibly be decoupled from the $\text{tr}(\Sigma^{1/2})/(\alpha\varepsilon)$ term. This is an artifact of the Gaussian noise added for privacy and can possibly be avoided using mean estimators based on the exponential mechanism, as in the spherical Gaussian case [13, 3, 35], but the volumetric arguments involved in their analysis incur factors dependent on $d$, which seem hard to overcome.

A more interesting direction for future work is the case of unknown covariance. We can determine special cases where the decay of $\Sigma$ allows us to achieve the optimal rate of Theorem 1.1 with unknown diagonal covariance. What is the appropriate norm in which one needs to learn $\Sigma$ for the current known-covariance approach to be accurate, and how many samples are needed for this task privately? More generally, the optimal sample complexity of mean estimation in the unknown (even diagonal) covariance case for anisotropic distributions (possibly achieved by an algorithm which doesn't follow the same structure) is an open question.

## Acknowledgments and Disclosure of Funding

We thank NeurIPS reviewers for suggestions on improving the clarity of this manuscript. This work was done while both LZ and YD were postdoctoral fellows in the Simons Institute for the Theory

of Computing, funded by FODSI. We also wish to acknowledge funding from the European Union (ERC-2022-SYG-OCEAN-101071601). Views and opinions expressed are however those of the author(s) only and do not necessarily reflect those of the European Union or the European Research Council Executive Agency. Neither the European Union nor the granting authority can be held responsible for them.

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

# A    Standard DP mechanisms and properties

**Definition A.1** (Laplace distribution). *For $v \geq 0$, let $\mathrm{Lap}(v)$ denote the Laplace distribution over $\mathbb{R}$, which has probability density function $p(z) = \frac{1}{2\sigma} e^{-|z|/v}$. From the CDF of the Laplace distribution, we get that $\Pr_{z \sim \mathrm{Lap}(v)}[z \geq v \log(1/2\beta)] = \beta$.*

**Definition A.2** (Laplace Mechanism, [25]). *Let $f : \mathcal{X}^* \to \mathbb{R}$, data set $X$ over $\mathcal{X}$, and privacy parameter $\varepsilon$. The* Laplace Mechanism *returns*

$$\tilde{f}(X) = f(X) + \mathrm{Lap}(v), \text{ where } v = \Delta_f / \varepsilon$$

*and $\Delta_f = \max_{X \sim X'} |f(X) - f(X')|$.*

**Lemma A.3** ([25]). *The Laplace Mechanism is $\varepsilon$-differentially private.*

**Definition A.4** (Gaussian Mechanism, [25]). *Let $f : \mathcal{X}^* \to \mathbb{R}^d$, data set $X$ over $\mathcal{X}$, and privacy parameters $\varepsilon, \delta$. The* Gaussian Mechanism *returns*

$$\tilde{f}(X) = f(X) + \mathcal{N}(0, v^2 I_d), \text{ where } v = \Delta_f \sqrt{2\log(1.25/\delta)}/\varepsilon$$

*and $\Delta_f = \max_{X \sim X'} \|f(X) - f(X')\|_2$ is the* global $\ell_2$-sensitivity *of $f$.*

**Lemma A.5** ([25]). *The Gaussian Mechanism is $(\varepsilon, \delta)$-differentially private.*

Differential privacy is maintained under post-processing and degrades mildly under composition.

**Lemma A.6** (Composition, [25, 27, 38]). *Let $M$ be an adaptive composition of $M_1, \ldots, M_T$, that is, on input $X$, $M(X) := M_T(X, M_{T-1}(X, \ldots, M_2(X, M_1(X))))$. Then*

1. *(Basic composition) If $M_1, \ldots, M_T$ are $(\varepsilon_1, \delta_1), \ldots, (\varepsilon_T, \delta_T)$-differentially private respectively, then $M$ is $(\varepsilon, \delta)$-differentially private for $\varepsilon = \sum_{t=1}^T \varepsilon_t$ and $\delta = \sum_{t=1}^T \delta_t$.*

2. *(Advanced composition) Let $\varepsilon_t > 0$, $\delta_t \in [0, 1]$ for $t \in \{1, \ldots, T\}$, and $\tilde{\delta} \in [0, 1]$. If $M_1, \ldots, M_T$ are $(\varepsilon_1, \delta_1), \ldots, (\varepsilon_T, \delta_T)$-differentially private respectively, then $M$ is $(\tilde{\varepsilon}_{\tilde{\delta}}, \tilde{\delta} + \sum_{t=1}^T \delta_t)$-differentially private where $\tilde{\varepsilon}_{\tilde{\delta}}$ is given by:*

$$\tilde{\varepsilon}_{\tilde{\delta}} = \sum_{\ell=1}^k \frac{(e^{\varepsilon_\ell} - 1)\varepsilon_\ell}{e^{\varepsilon_\ell} + 1} + \sqrt{\sum_{\ell=1}^k \varepsilon_\ell^2 \log\left(\frac{1}{\tilde{\delta}}\right)}.$$

**Fact A.7** (Fact 2.17 [63] reduced to pure DP). *Let $Y \approx_\varepsilon Y'$ random variables over $\mathcal{Y}$ and let the event $E \subseteq \mathcal{Y}$ be such that $\Pr[Y \in E], \Pr[Y' \in E] \geq q$. Then $Y_{|E} \approx_{\varepsilon/q} Y'_{|E}$.*

# B    Omitted details of Section 3

We state here again the general theorem which holds for any $M$.

**Theorem B.1.** *Let $\varepsilon \in (0, 10), \delta \in (0, 1), \alpha > 0, \beta \in (0, 1)$.[4] Algorithm 1 is $(\varepsilon, \delta)$-differentially private. Let $X$ be a data set of size $n$, drawn from a subgaussian distribution with covariance-proxy $\Sigma$ and mean $\mu$. Given $M = \Sigma$, $\lambda \geq \sqrt{2 \operatorname{tr}(M^{-1/4} \Sigma M^{-1/4})} + 2\sqrt{2\|M^{-1/4} \Sigma M^{-1/4}\|_2 \log(\frac{n}{\beta})}$, with probability at least $1 - \beta$, Algorithm 1 returns $\hat{\mu}$ such that $\|\hat{\mu} - \mu\|_2 \leq \alpha$, as long as*

$$n \geq C \left( \frac{\operatorname{tr}(\Sigma) + \|\Sigma\|_2 \log \frac{1}{\beta}}{\alpha^2} + \lambda \left( \sqrt{\operatorname{tr}(M^{1/2})} + \sqrt{\|M^{1/2}\|_2 \log \frac{1}{\beta}} \right) \frac{\sqrt{\log \frac{1}{\delta}}}{\alpha \varepsilon} + \frac{\log \frac{1}{\delta\beta}}{\varepsilon} \right),$$

$$(5)$$

---

[4]We require $\varepsilon$ to be smaller than some constant, due to approximations we take in the privacy analysis. The theorem may hold for $\varepsilon > 10$ but we did not optimize the choice of constant, as this range is already wide.

*for some universal constant C. If $\Sigma$ is known, choosing $M = \Sigma$ and substituting $\lambda$, the sample complexity becomes*

$$n \geq C\left(\frac{\mathrm{tr}(\Sigma) + \|\Sigma\|_2 \log\frac{1}{\beta}}{\alpha^2} + \frac{\mathrm{tr}(\Sigma^{1/2})\sqrt{\log\frac{1}{\delta}}}{\alpha\varepsilon}\right.$$
$$\left. + \frac{\sqrt{\|\Sigma\|_2 \log\frac{1}{\delta}} \log\frac{1}{\beta}}{\alpha\varepsilon} \log\left(\frac{\|\Sigma\|_2 \log\frac{1}{\delta} \log\frac{1}{\beta}}{\alpha\varepsilon}\right) + \frac{\log\frac{1}{\delta\beta}}{\varepsilon}\right). \tag{6}$$

Although the theorem holds for any $M, \lambda$, choosing $M = \Sigma$ gives us the optimal bound.[5]

*Remark 2.* If $M$ is such that $\Sigma \preceq M$, we may assume without loss of generality that $M$ is invertible. Indeed, if this is not the case, then we know that the distribution of the data is supported on a lower-dimensional subspace along with its mean $\mu$. Using $M$, we can project onto this subspace. In this context, we can refocus our analysis on the scenario where $M$ is an invertible matrix.

**Computational complexity**   We note that Algorithm 1 has time complexity $O(n^2)$, which can be further improved to $O(n \log n)$ as in [63].

The guarantees of Theorem B.1 follow by combining Theorem B.3 and Theorem B.5 below and re-scaling parameters $\varepsilon, \delta, \beta$ with appropriate constants.

### B.1   Accuracy analysis

We start with the accuracy analysis. We first prove that for subgaussian data sets, all data points pass the filter with high probability.

**Lemma B.2.** *Let $X = (X^{(1)}, \ldots, X^{(n)})$ be a data set drawn from a subgaussian distribution with covariance proxy $\Sigma$. Let $\beta \in (0,1)$, $M$ invertible matrix and some $\lambda \geq \sqrt{2\,\mathrm{tr}(M^{-1/4}\Sigma M^{-1/4})} + 2\sqrt{2\|M^{-1/4}\Sigma M^{-1/4}\|_2 \log(n/\beta)}$ given as inputs to Algorithm 1. Then the* BasicFilter *procedure outputs $C = X$, with probability $1 - \beta$.*

*Proof.* It suffices to show that in BasicFilter we have $p_j = 1$ for $j \in [n]$ (so that $v_j = 1$ and thus $C = X$). For each $j, k \in [n]$, we want to show $\mathrm{dist}_{M,\lambda}(X^{(j)}, X^{(k)}) = 1$ with probability at least $1 - \beta/n^2$. For $j = k$, it is trivial. What is left is to show for $j \neq k$,

$$\|M^{-1/4}(X^{(j)} - X^{(k)})\|_2 \leq \sqrt{2\,\mathrm{tr}(M^{-1/4}\Sigma M^{-1/4})} + 2\sqrt{2\|M^{-1/4}\Sigma M^{-1/4}\|_2 \log(n/\beta)}, \tag{7}$$

with probability at least $1 - \beta/n^2$. First observe that $-X^{(k)}$ is a subgaussian vector independent of $X^{(j)}$ with mean $-\mu$ and covariance proxy $\Sigma$. Hence, $X^{(j)} - X^{(k)}$ is a subgaussian vector with mean 0 and covariance proxy $2\Sigma$, and so $M^{-1/4}(X^{(j)} - X^{(k)})$ is subgaussian with mean zero and covariance proxy $2M^{-1/4}\Sigma M^{-1/4}$. By Lemma 2.7, with probability $1 - \beta/n^2$,

$$\|M^{-1/4}(X^{(j)} - X^{(k)})\|_2 \leq \sqrt{2\,\mathrm{tr}(M^{-1/4}\Sigma M^{-1/4})} + 2\sqrt{2\|M^{-1/4}\Sigma M^{-1/4}\|_2 \log(n/\beta)}.$$

Then we union bound all $n(n-1)$ pairs of $j, k \in [n], j \neq k$ such that Eq. (7) holds with probability of at least $1 - \beta$. $\qquad\square$

---

[5] For intuition, consider adding noise $\mathcal{N}(0, c_i^2)$ to each coordinate $i$. (It is clear that the privacy budget should be distributed unevenly among coordinates.) We would like to minimize the $\ell_2$ norm of this noise, which is approximately $\sum_{i=1}^d c_i^2$. The average sensitivity of each coordinate of a Gaussian is $\sigma_i$, so to achieve total privacy loss $\varepsilon$, by advanced composition, we require the $c_i$'s to satisfy $\sum_{i=1}^d \sigma_i^2/c_i^2 \leq \varepsilon^2$. Solving this optimization problem, we find that $c_i \propto \sqrt{\|\boldsymbol{\sigma}\|_1 \sigma_i}$, which corresponds to noise $\mathcal{N}(0, \Delta^2 \Sigma^{1/2})$, for $\Delta^2 \approx \frac{\|\boldsymbol{\sigma}\|_1}{\varepsilon^2 n^2}$. The same reasoning can be applied to the case of the Mahalanobis error metric [13] where adding noise $\mathcal{N}(0, \Delta_M^2 \Sigma)$ for $\Delta_M^2 \approx \frac{d}{\varepsilon^2 n^2}$ gives us the optimal bound. Here the minimization objective is roughly $\Sigma_{i=1}^d c_i^2/\sigma_i^2$ so the optimal solution requires $c_i \propto \sigma_i$.

**Theorem B.3.** *Let $\varepsilon > 0, \delta \in (0,1), \alpha > 0, \beta \in (0,1)$. Suppose $n \geq 2\log(1/\delta\beta)/\varepsilon$. Let $X = (X^{(1)}, \ldots, X^{(n)})$ be a data set drawn from a subgaussian distribution with mean $\mu$ and covariance proxy $\Sigma$. Then, given invertible matrix $M$ and $\lambda \geq \sqrt{2\operatorname{tr}(M^{-1/4}\Sigma M^{-1/4})} + 2\sqrt{2\|M^{-1/4}\Sigma M^{-1/4}\|_2 \log(n/\beta)}$, Algorithm 1, with probability $1 - \frac{7}{2}\beta$, returns $\hat{\mu}$ such that*

$$\|\hat{\mu} - \mu\|_2 \leq \frac{\sqrt{\operatorname{tr}(\Sigma)}}{\sqrt{n}} + \frac{\sqrt{2\|\Sigma\|_2 \log(1/\beta)}}{\sqrt{n}}$$
$$+ \frac{4\sqrt{2\log(1.25/\delta)}\lambda}{\varepsilon n} \left( \sqrt{\operatorname{tr}(M^{1/2})} + \sqrt{2\|M^{1/2}\|_2 \log(1/\beta)} \right).$$

*Proof.* Let $\mu_C, \mu_X$ be the sample mean of $C$ and $X$, respectively. By *the triangle* inequality, we decompose it into

$$\|\hat{\mu} - \mu\|_2 \leq \|\hat{\mu} - \mu_C\|_2 + \|\mu_C - \mu_X\|_2 + \|\mu_X - \mu\|_2. \tag{8}$$

By Lemma B.2, $C = X$ with probability $1 - \beta$. Condition on this event for the rest of the proof. Then, $\hat{n}_C = n - \frac{\log(1/\delta)}{\varepsilon} + z$ satisfies $\hat{n}_C \geq 0.5n > 0$ with probability $1 - \beta/2$ because

$$\Pr\left[z < \frac{\log(1/\delta)}{\varepsilon} - 0.5n\right] \leq \Pr\left[z < \frac{\log(1/\delta)}{\varepsilon} - \frac{\log(1/\delta\beta)}{\varepsilon}\right] = \Pr\left[z < -\frac{\log(1/\beta)}{\varepsilon}\right] = \frac{1}{2}\beta$$

by Definition A.1 and our assumption that $n \geq 2\log(1/\delta\beta)/\varepsilon$. Conditioning on this assumption, we do not abort and with probability $1 - \beta$, by Lemma 2.7,

$$\|\hat{\mu} - \mu_C\|_2 = \|\eta\|_2 \leq \frac{4\sqrt{2\log(1.25/\delta)}\lambda}{\varepsilon n} \left( \sqrt{\operatorname{tr}(M^{1/2})} + \sqrt{2\|M^{1/2}\|_2 \log(1/\beta)} \right).$$

Again, by Lemma 2.7, with probability $1 - \beta$,

$$\|\mu_X - \mu\|_2 = \left\| \frac{1}{n} \sum_{j=1}^{n} X^{(j)} - \mu \right\|_2 \leq \frac{\sqrt{\operatorname{tr}(\Sigma)}}{\sqrt{n}} + \frac{\sqrt{2\|\Sigma\|_2 \log(1/\beta)}}{\sqrt{n}}.$$

Moreover, since $C = X$, it holds that $\mu_X = \mu_C$. Combining these results into Eq. (8), the algorithm does not abort and we retrieve the stated error bound, with probability $1 - \frac{7}{2}\beta$. $\square$

### B.2 Privacy analysis

We now move to the privacy analysis.

**Lemma B.4.** *In Algorithm 1, $\hat{n}_C < |C|$, with probability $1 - \delta/2$.*

*Proof.* It follows that by Definition A.1,

$$\Pr\left[\hat{n}_C \geq |C|\right] = \Pr\left[|C| - \log(1/\delta)/\varepsilon + z \geq |C|\right] = \Pr\left[z \geq \log(1/\delta)/\varepsilon\right] = \frac{1}{2}\delta.$$

Note that this holds regardless of whether $C$ is $\operatorname{dist}_{\Sigma,\beta}$-friendly or whether $X$ is subgaussian. $\square$

The privacy analysis follows the steps of [63, Claim 3.4].

**Theorem B.5.** *Let $\varepsilon \in (0, 1/2), \delta \in (0, 1/2)$. For any input parameters $M, \lambda$, Algorithm 1 satisfies $(21\varepsilon, e^{10}\delta)$-DP.*

*Proof.* It suffices show that lines 4-7 of Algorithm 1 are $\operatorname{dist}_{\Sigma,\beta}$-friendly $(\varepsilon', \delta')$-DP, such that by Theorem 2.5, Algorithm 1 is $(2(e^{\varepsilon'} - 1)\varepsilon', 2e^{\varepsilon' + 2(e^{\varepsilon'} - 1)}\delta')$-DP.

Denote lines 4-7 of Algorithm 1 as algorithm $\mathcal{A}$. Consider neighboring inputs $X, X'$ such that $X \cup X'$ is $\operatorname{dist}_{\Sigma,\beta}$-friendly. Assume without loss of generality $X' = X \setminus X^{(j)}$ so that $|X'| = |X| - 1$. Let $\mathcal{A}(X), \mathcal{A}(X')$ represent the outputs of two independent executions of $\mathcal{A}$ and let $\widehat{N}_X, \widehat{N}_{X'}$ be the random variable in line 4 of the algorithm. We want to show $\mathcal{A}(X) \approx_{\varepsilon', \delta'} \mathcal{A}(X')$.

Note that $|X| > 0$. If $|X'| = 0$, then $|X| = 1$ and $\Pr[\mathcal{A}(X') = \perp] = 1$. We then show that $\Pr[\mathcal{A}(X) = \perp] \geq 1 - e^\varepsilon \delta/2$. This holds since by Definition A.1,

$$\Pr[\hat{N}_X \leq 0] = \Pr[z \leq \log(1/\delta)/\varepsilon - 1] = \Pr[z \leq \log(1/(e^\varepsilon \delta))/\varepsilon] = 1 - \frac{e^\varepsilon \delta}{2}.$$

Therefore, in this case, $\mathcal{A}(X) \approx_{0,e^\varepsilon \delta/2} \mathcal{A}(X')$. That is, if $\varepsilon \leq 1/2$, $\mathcal{A}$ is $(0, \delta)$-DP.

Now consider $|X'| > 0$. By Lemma B.4, We know $\Pr[\hat{N}_X < |X|] = \Pr[\hat{N}_{X'} < |X'|] = 1 - \delta/2$. Hence, $\Pr[\hat{N}_{X'} < |X|] = \Pr[\hat{N}_{X'} < |X'| + 1] \geq 1 - \delta/2$. Then what is left is to compare $\mathcal{A}(X)|_{\hat{N}_X < |X|}, \mathcal{A}(X')|_{\hat{N}_{X'} < |X|}$.

By Lemma A.3, $\hat{N}_X \approx_{\varepsilon,0} \hat{N}_{X'}$ as $|X| - |X'| = 1$. By Fact A.7, $\hat{N}_X|_{\hat{N}_X < |X|} \approx_{\varepsilon/(1-\delta/2),0} \hat{N}_{X'}|_{\hat{N}_{X'} < |X|}$. In order to perform composition by Lemma A.6, we now show that for each fixed $\hat{n} < |X|$, $\mathcal{A}(X)|_{\hat{N}_X = \hat{n}} \approx_{\varepsilon,\delta} \mathcal{A}(X')|_{\hat{N}_{X'} = \hat{n}}$ as follows:

Choose $\hat{n} < |X|$. If $\hat{n} \leq 0$, then $\mathcal{A}(X)|_{\widehat{N}_X = \hat{n}} = \mathcal{A}(X')|_{\widehat{N}_X = \hat{n}} = \perp$ and we are done. If $0 < \hat{n} < |X|$, it suffices to show $\mathcal{N}(\frac{1}{|X|} \sum_{i=1}^{|X|} X^{(i)}, v^2 M^{1/2}) \approx_{\varepsilon,\delta} \mathcal{N}(\frac{1}{|X|-1} \sum_{i=1,i\neq j}^{|X|} X^{(i)}, v^2 M^{1/2})$, where $v^2 = \frac{8\log(1.25/\delta)\lambda^2}{\varepsilon^2 \hat{n}^2}$, which, by post-processing, is equivalent to

$$\mathcal{N}\left(M^{-1/4} \frac{1}{|X|} \sum_{i=1}^{|X|} X^{(i)}, v^2 I_d\right) \approx_{\varepsilon,\delta} \mathcal{N}\left(M^{-1/4} \frac{1}{|X|-1} \sum_{i=1,i\neq j}^{|X|} X^{(i)}, v^2 I_d\right). \tag{9}$$

Define vector $D = \frac{1}{|X|} \sum_{i=1}^{|X|} X^{(i)} - \frac{1}{|X|-1} \sum_{i=1,i\neq j}^{|X|} X^{(i)}$. As $X \cup X'$ is $\text{dist}_{\Sigma,\beta}$-friendly, for every $i \in [|X|] \setminus \{j\}$, there exists some $Y^{(i)} \in \mathbb{R}^d$ such that $\text{dist}_{\Sigma,\beta}(X^{(i)}, Y^{(i)}) = \text{dist}_{\Sigma,\beta}(X^{(j)}, Y^{(i)}) = 1$. We have

$$\|M^{-1/4} D\|_2 = \frac{1}{|X|(|X|-1)} \left\| M^{-1/4} \left( \left(\sum_{i=1,i\neq j}^{|X|} X^{(i)}\right) - X^{(j)}(|X|-1)\right) \right\|_2$$

$$\leq \frac{1}{|X|(|X|-1)} \sum_{i=1,i\neq j}^{|X|} \left( \left\| M^{-1/4}\left(X^{(i)} - Y^{(i)}\right) \right\|_2 + \left\| M^{-1/4}\left(Y^{(i)} - X^{(j)}\right) \right\|_2 \right)$$

(by triangle inequality)

$$\leq \frac{1}{|X|(|X|-1)} \sum_{i=1,i\neq j}^{|X|} 2\lambda = \frac{2\lambda}{|X|} \leq \frac{2\lambda}{\hat{n}}.$$

(by $\text{dist}_{\Sigma,\beta}$-friendly assumption and since $0 < \hat{n} < |X|$)

We know Equation (9) holds by applying the guarantees of the Gaussian mechanism (Lemma A.5) where we set $f(X) = M^{-1/4} \frac{1}{|X|} \sum_{i=1}^{|X|} X^{(i)}$ and $\Delta_f = 2\lambda/\hat{n}$.

Combining these results, we have $\mathcal{A}$ is $(\varepsilon + \frac{\varepsilon}{1-\delta/2}, \delta e^{\varepsilon/(1-\delta/2)} + \frac{\delta}{2})$-DP in this case. For $\varepsilon \leq 1/2, \delta \leq 1/2$, this becomes at most $(3\varepsilon, 2\delta)$-DP.

Therefore, overall, by Theorem 2.5, Algorithm 1 is $(2(e^{\varepsilon'} - 1)\varepsilon', 2e^{\varepsilon' + 2(e^{\varepsilon'} - 1)}\delta')$-DP with $\varepsilon' = 3\varepsilon, \delta' = 2\delta$. So for $\varepsilon \leq 1/2$, the algorithm is $(21\varepsilon, e^{10}\delta)$-DP overall. $\qquad\square$

## C  Lower bounds

### C.1  Dimension-dependent lower bound under pure DP

The so-called *packing* lower bound technique [32, 10] implies a lower bound on the order of $d$ for the number of samples required by any *pure* DP algorithm learning the mean of a Gaussian distribution, even in the anisotropic case we consider in this paper.

There exist several statements in prior works which establish the lower bound for learning a Gaussian distribution with known covariance in TV distance, which is equivalent to learning the mean in Mahalanobis distance, or to learning the mean in $\ell_2$ norm in the isotropic case [18, Lemma 5.1]. It is trivial to observe that the dependence on the dimension $d$ persists in the anisotropic case, yet we include the proof here for completeness.

**Theorem C.1.** *For any $\alpha < R/2$, any $\varepsilon$-DP algorithm which estimates the mean $\mu \in \mathcal{B}^d(R)$ of a Gaussian distribution with known covariance $\Sigma$, up to accuracy $\alpha$ in $\ell_2$ norm with probability $9/10$, requires $n \geq \frac{d\log(R/2\alpha)}{\varepsilon}$ samples.*

*Proof.* Consider a $2\alpha$-packing of the $d$-dimensional $R$-radius ball, denoted by $\mathcal{P}_{2\alpha} \subset \mathcal{B}^d(R)$. That is, $\forall u, v \in \mathcal{P}$, $\|u - v\|_2 > 2\alpha$, so that the balls with centers $u, v$ and radius $\alpha$ are disjoint: $\mathcal{B}^d(u, \alpha) \cap \mathcal{B}^d(v, \alpha) = \emptyset$. We consider the family of Gaussian distributions $\{\mathcal{N}(u, \Sigma)\}_{u \in \mathcal{P}_{2\alpha}}$. Suppose $\mathcal{A}$ is an $\varepsilon$-DP algorithm with the stated accuracy requirement. This implies that $\forall u \in \mathcal{P}_{2\alpha}$:

$$\Pr_{\mathcal{A}, X \sim \mathcal{N}(u, \Sigma)^n}[\mathcal{A}(X) \in \mathcal{B}^d(u, \alpha)] \geq 9/10. \tag{10}$$

At the same time, for any pair of samples $X, X_0$ of size $n$, and any measurable set $B \subset \mathrm{range}(\mathcal{A})$, by the privacy guarantee, $\Pr_{\mathcal{A}}[\mathcal{A}(X) \in B] \leq e^{\varepsilon n} \Pr_{\mathcal{A}}[\mathcal{A}(X_0) \in B]$. This implies specifically that for $u_0, u \in \mathcal{P}_{2\alpha}$,

$$\Pr_{\mathcal{A}, X \sim \mathcal{N}(u, \Sigma)^n}[\mathcal{A}(X) \in B] \leq e^{\varepsilon n} \Pr_{\mathcal{A}, X_0 \sim \mathcal{N}(u_0, \Sigma)^n}[\mathcal{A}(X_0) \in B]. \tag{11}$$

We have

$$
\begin{aligned}
1 &\geq \Pr_{\mathcal{A}, X_0 \sim \mathcal{N}(u_0, \Sigma)^n}\Big[\mathcal{A}(X_0) \in \bigcup_{u \in \mathcal{P}_{2\alpha}} \mathcal{B}^d(u, \alpha)\Big] \\
&= \sum_{u \in \mathcal{P}_{2\alpha}} \Pr_{\mathcal{A}, X_0 \sim \mathcal{N}(u_0, \Sigma)^n}[\mathcal{A}(X_0) \in \mathcal{B}^d(u, \alpha)] && (\{\mathcal{B}^d(u, \alpha)\}_{u \in \mathcal{P}_{2\alpha}} \text{ disjoint}) \\
&\geq \sum_{u \in \mathcal{P}_{2\alpha}} e^{-\varepsilon n} \Pr_{\mathcal{A}, X \sim \mathcal{N}(u, \Sigma)^n}[\mathcal{A}(X) \in \mathcal{B}^d(u, \alpha)] && (\text{by Eq. (11)}) \\
&\geq |\mathcal{P}_{2\alpha}| e^{-\varepsilon n} \cdot \frac{9}{10}. && (\text{by Eq. (10)})
\end{aligned}
$$

We conclude that $n \geq \frac{\log|\mathcal{P}_{2\alpha}|}{\varepsilon}$. Since $|\mathcal{P}_{2\alpha}| \geq \left(\frac{R}{\alpha}\right)^d$, it follows that $n \geq \frac{d\log(R/2\alpha)}{\varepsilon}$. $\qquad\square$

This lower bound makes $\varepsilon$-DP prohibitive for the regime we consider in our setting. To compare with our upper bounds for $(\varepsilon, \delta)$-DP, suppose that we want to learn $\mu$ with accuracy $c\sigma_1 < R$, where $c > 0$ is a small constant. Then our main result implies that this is achievable with $n \leq C\frac{\|\boldsymbol{\sigma}\|_1}{\varepsilon\sigma_1}$ samples for some constant $C > 0$, whereas under $\varepsilon$-DP, we would need at least $n \geq \frac{d}{\varepsilon} \gg \frac{\|\boldsymbol{\sigma}\|_1}{\varepsilon\sigma_1}$ for the regime we consider in this paper.

## C.2 Lower bound for approximate DP

The so-called *tracing* or *fingerprinting* lower-bound technique [15, 60, 17, 61, 28] is the main technique used to yield lower bounds for mean estimation under $(\varepsilon, \delta)$-DP. Kamath et al. [39, 40] apply it to give lower bounds for the problem of learning a Gaussian in TV distance (which is equivalent to learning the Gaussian in Mahalanobis distance for the known covariance case, or to the isotropic case).

**Theorem C.2** (Theorem 6.5 [39]). *If $\mathcal{A}: \mathbb{R}^{d \times n} \to [-R\sigma, R\sigma]^d$ is $(\varepsilon, \delta)$-DP for $\delta = \tilde{O}(\frac{\sqrt{d}}{Rn})$, and for every Gaussian distribution with mean $\mu \in [-R\sigma, R\sigma]^d$ and known covariance matrix $\sigma^2 I_d$, with probability $2/3$, $\|\mathcal{A}(X) - \mu\| \leq \alpha \leq \sqrt{d}\sigma R/3$, then $n \geq \frac{d\sigma}{24\alpha\varepsilon \log(dR)}$.*

Following exactly the same steps as the proof of the theorem under the slightly more general case of known covariance $\Sigma = \mathrm{diag}(\boldsymbol{\sigma}^2)$ gives us a weak lower bound for our setting, on the order of $n \geq \frac{\|\boldsymbol{\sigma}\|_2^2}{24\varepsilon\alpha\sigma_1^2 \log(dR)}$.

However, a more careful application of the same theorem directly gives us the following stronger lower bound, which implies that our algorithm for the known covariance case is near-optimal.

**Theorem C.3.** *If $\mathcal{A} : \mathbb{R}^{d \times n} \to \mathbb{R}^d$ is $(\varepsilon, \delta)$-DP for $\delta = O((n\sqrt{\log(n)})^{-1})$, and for every Gaussian distribution with mean $\mu \in [-1, 1]^d$ and known covariance proxy $\Sigma = \mathrm{diag}(\boldsymbol{\sigma}^2)$, with probability $2/3$, $\|\mathcal{A}(X) - \mu\| \leq \alpha = O(\|\boldsymbol{\sigma}\|_1 / \log(d))$, then $n = \Omega\left(\frac{\|\boldsymbol{\sigma}\|_1}{\alpha\varepsilon \log^2(d)}\right)$.*

*Proof.* Assume w.l.o.g. that $\sigma_1^2 \geq \ldots \geq \sigma_d^2$. Consider a partition of the set of coordinates $[d]$ into buckets $S_k = \{i \in [d] : \sigma_i \in (\frac{\sigma_1}{2^k}, \frac{\sigma_1}{2^{k-1}}]\}$, $\forall k \in [\log(d)]$ and $S_{\log(d)+1} = [d] \setminus \bigcup_{k \in [\log(d)]} S_k$. We have that $\sum_{k=1}^{\log(d)+1} \sum_{i \in S_k} \sigma_i = \|\boldsymbol{\sigma}\|_1$. Consider the bucket $S_m$ which contributes the most to this sum, that is $m = \arg\max \sum_{i \in S_m} \sigma_i$. Let $\sigma_{S_m} = \max\{\sigma_i : i \in S_m\}$. It must be that

$$|S_m| \geq \frac{\|\boldsymbol{\sigma}\|_1}{(\log(d)+1)\sigma_{S_m}}.$$

Otherwise, $\|\boldsymbol{\sigma}\|_1 = \sum_{k=1}^{\log(d)+1} \sum_{i \in S_k} \sigma_i \leq (\log(d)+1)|S_m|\sigma_{S_m} < \|\boldsymbol{\sigma}\|_1$, which is a contradiction. All the variances of the coordinates in $S_m$ are within a factor of two from $\sigma_{S_m}$. We apply Theorem C.2 to the $|S_m|$-dimensional Gaussian with $R = 1$. Consider the Gaussian distribution with mean $\mu_{S_m} \in [-1, 1]^{|S_m|}$ and known covariance matrix $\sigma_{S_m}^2 I_d$. We have that any $(\varepsilon, \delta)$-DP algorithm for $\delta = O\left(\frac{1}{n\sqrt{\log(n)}}\right)$ which returns, with probability $2/3$, an estimate $\hat{\mu}_{S_m}$ with error $\|\hat{\mu}_{S_m} - \mu_{S_m}\|_2 \leq \alpha \leq \sqrt{|S_m|}\sigma_{S_m}/3$, requires

$$n \geq \frac{|S_m|\sigma_{S_m}}{24\alpha\varepsilon \log(d)} \geq \frac{\|\boldsymbol{\sigma}\|_1}{48\alpha\varepsilon \log^2(d)} \tag{12}$$

samples.

Now assume that there exists $(\varepsilon, \delta)$-DP algorithm $\mathcal{A} : \mathbb{R}^{d \times n} \to \mathbb{R}^d$ for $\delta = O\left(\frac{1}{n\sqrt{\log(n)}}\right)$, such that, for every Gaussian distribution with mean $\mu \in [-1, 1]^d$ and known covariance proxy $\Sigma = \mathrm{diag}(\boldsymbol{\sigma}^2)$, with probability $2/3$, $\|\mathcal{A}(X) - \mu\| \leq \alpha \leq \frac{\|\boldsymbol{\sigma}\|_1}{3(\log(d)+1)}$. Restricting the output $\mathcal{A}(X)$ to the coordinates in $S_m$, would give us a mean estimate for $S_m$ with error at most $\alpha$. Combined with Eq. (12), this completes the proof of the theorem. $\qquad\square$

## D  Omitted details of Section 4

We state the main theorem in more detail.

**Theorem D.1.** *Let parameters $\varepsilon, \delta \in (0, 1)$. Let $X \sim \mathcal{N}(\mu, \Sigma)^n$ with unknown covariance $\Sigma$. There exists an $(\varepsilon, \delta)$-differentially private algorithm which, with probability $1 - \beta$, returns an estimate $\hat{\mu}$ such that $\|\hat{\mu} - \mu\|_2 \leq \alpha$, as long as*

$$n = \Omega\left(\log^2(d) + \frac{\log(d)\log(\frac{1}{\delta\beta})\sqrt{\log\frac{1}{\delta}}}{\varepsilon}\right),$$

$$n = \Omega\left(\frac{\mathrm{tr}(\Sigma) + \|\Sigma\|_2 \log\frac{1}{\beta}}{\alpha^2}\right), \tag{13}$$

$$n = \tilde{\Omega}\left(\frac{\sum_{i=1}^d \Sigma_{ii}^{1/2} \sqrt{\log\frac{1}{\delta}}}{\alpha\varepsilon}\right), \tag{14}$$

*and*

$$n = \tilde{\Omega}\left(\frac{d^{1/4}\sqrt{\sum_{i=1}^d \Sigma_{ii}^{1/2}} \log^{5/4}(\frac{1}{\delta}) \log(d)}{\sqrt{\alpha}\varepsilon}\right), \tag{15}$$

*where the symbol $\tilde{\Omega}$ hides multiplicative logarithmic factors in $1/\beta$ and the term in parentheses.*

Recall the definition of set $V$. Within each group $j$, for each coordinate $i$, we compute an estimate $V_i^{(j)}$ for $\Sigma_{ii}$ as follows:

$$V_i^{(j)} = \frac{1}{2\ell} \sum_{r=1}^{\ell} \left( X_i^{(j,2r-1)} - X_i^{(j,2r)} \right)^2 \ . \tag{16}$$

We show that the variance estimates are valid in the subsequent sections.

**Lemma D.2.** *Let $X^{(1)}, \ldots, X^{(n)}$ be $d$-dimensional i.i.d. samples from $\mathcal{N}(\mu, \Sigma)$. Let $\{V_i^{(j)}\}_{j \in [m], i \in [d]}$ be the estimates defined in Eq. (16). Then, there exist universal constants $C, C' > 1$ such that if $\ell \geq C \log d$ and $m \geq C' \log(1/\beta)$, with probability at least $1 - \beta$, the set $V$ of estimates is valid.*

Next, we use the estimates $V_i^{(j)}$ as inputs to multiple procedures. We introduce the following estimation tasks.

**Definition D.3.** *For a covariance matrix $\Sigma \in \mathbb{R}^{d \times d}$ consider the following:*

1. *$k$-**th largest variance:** Approximate the $k$-th largest value among the diagonal of $\Sigma$, namely, the $k$-th largest value among $(\Sigma_{11}, \ldots, \Sigma_{dd})$.*

2. ***Sum of variances:*** *given a subset $I \subseteq [d]$, approximate the sum $\sum_{i \in I} \Sigma_{ii}$.*

We have the following algorithms for these tasks. The proofs of Lemmas D.4, D.5, and D.6 are in the subsequent subsections.

**Lemma D.4.** *Let $\varepsilon, \delta, \beta \in (0, 1/2)$. There exists an algorithm $\mathrm{FindKthLargestVariance}_{\varepsilon, \delta}$, which receives variance estimates $V^{(1)}, \ldots, V^{(m)} \in \mathbb{R}^d$ and an integer $k \in [d]$, and satisfies the following, provided that*

$$m \geq \Omega\left( \frac{1}{\varepsilon} \log \frac{1}{\delta\beta} \right) \ .$$

- *Privacy: $\mathrm{FindKthLargestVariance}_{\varepsilon, \delta}$ is $(\varepsilon, \delta)$-DP with respect to changing each input vector $V^{(j)}$.*

- *Accuracy: denote the $k$-th largest entry of $\{\Sigma_{11}, \ldots, \Sigma_{dd}\}$ by $Q$ and the algorithm's output by $\hat{Q}$. If the estimates $(V^{(1)}, \ldots, V^{(m)})$ are valid wrt $\Sigma$, then there exists a universal constant $C > 0$ such that with probability at least $1 - \beta$,*

$$Q/8 \leq \hat{Q} \leq 8Q \ .$$

**Lemma D.5.** *Let $\varepsilon, \delta, \beta \in (0, 1)$. There exists an algorithm $\mathrm{VarianceSum}_{\varepsilon, \delta}$, which receives variance estimates $V^{(1)}, \ldots, V^{(m)} \in \mathbb{R}^d$ and a subset $I \subseteq [d]$. It has the exact same guarantees as $\mathrm{FindKthLargestVariance}$ from Lemma D.4, except that it provides an estimate for $\sum_{i \in I} \Sigma_{ii}$ instead of an estimate for the $k$-th largest diagonal entry of $\Sigma$.*

Assume for now that the estimates $V_i^{(j)}$ are valid. With these procedures at hand, we first compute $R$ such that (by rescaling) $Q/64 \leq R \leq Q$, where $Q$ is the $k$-th largest diagonal entry of $\Sigma$. Then, we call a procedure that finds $k$ entries $i$ such that $\Sigma_{ii} \geq R$. Its guarantees are listed below:

**Lemma D.6.** *Let $\varepsilon, \delta, \beta \in (0, 1)$. There exists an $(\varepsilon, \delta)$-DP algorithm $\mathrm{TopVar}_{\varepsilon, \delta}(V, R)$, such that, if $V$ is valid,*

$$m \geq \Omega\left( \sqrt{\frac{k \log(1/\delta)}{\varepsilon n}} \log \frac{d}{\beta} \right),$$

*and $|\{i \colon \Sigma_{ii} \geq R\}| \geq k$, then the algorithm outputs a set $I_{\mathrm{top}}$ of size $k$ such that for all $i \in I_{\mathrm{top}}$, $\Sigma_{ii} \geq R/4$.*

At the next step, we would like to find, up to a constant factor, the variances corresponding to these coordinates: the values $\Sigma_{ii}$ for $i \in I_{\mathrm{top}}$. We use the algorithm $\mathrm{VarianceSum}$ $k$ times, providing the sets $\{i\}$ for $i \in I_{\mathrm{top}}$. We obtain estimates $\hat{\Sigma}_{ii}$ that approximate $\Sigma_{ii}$ up to a constant factor.

Next, we estimate the mean $\mu$ in the coordinates $I_{\mathrm{top}}$, denoted $\mu_{\mathrm{top}}$, separately from $I_{\mathrm{bot}} := [d] \setminus I_{\mathrm{top}}$, denoted $\mu_{\mathrm{bot}}$: since we approximately know the variances in $I_{\mathrm{top}}$, we can obtain a better estimate.

Both for estimating $\mu_{\text{top}}$, and for estimating $\mu_{\text{bot}}$, we use $\text{Avg}_{M,\lambda,\varepsilon,\delta}$ (Algorithm 1 from Section 3) with appropriate choices of parameters $M, \lambda$. Recall that Algorithm 1 satisfies the guarantees of Theorem B.1.

For estimating $\mu_{\text{top}}$, we use $\text{Avg}_{M,\lambda,\varepsilon,\delta}$ for estimating the mean of a $k$-dimensional Gaussian, with input vectors restricted to coordinates $I_{\text{top}}$, $X_{I_{\text{top}}}^{(1)}, \ldots, X_{I_{\text{top}}}^{(n)}$, the $k \times k$-dimensional diagonal matrix $M$, with $M_{ii} = \hat{\Sigma}_{ii}$ (we assume that the rows and columns of $M$ are indexed by $I_{\text{top}}$), and $\lambda = O\left(\sqrt{\sum_{i \in I_{\text{top}}} \hat{\Sigma}_{ii}^{1/2} \log \frac{n}{\beta}}\right)$. Denote the output by $\hat{\mu}_{\text{top}}$. Theorem B.1 shows that with probability $1 - \beta$, $\|\hat{\mu}_{\text{top}} - \mu_{\text{top}}\|_2 \leq \alpha$, if

$$n \geq \tilde{\Omega}\left(\frac{\text{tr}(\Sigma)}{\alpha^2} + \frac{\sqrt{\log(1/\delta)} \sum_{i \in I_{\text{top}}} \Sigma_{ii}^{1/2}}{\alpha\varepsilon} + \frac{\log(1/\delta)}{\varepsilon}\right), \tag{17}$$

where $\tilde{\Omega}$ hides multiplicative logarithmic factors in $1/\beta$ and the second term.

For estimating $I_{\text{bot}}$, we do not know the variances. In order to perform the estimation, we first call the algorithm VarianceSum to provide an estimate $\hat{S}_{\text{bot}}$ such that $\frac{1}{C} \sum_{i \in I_{\text{bot}}} \Sigma_{ii} \leq \hat{S}_{\text{bot}} \leq C \sum_{i \in I_{\text{bot}}} \Sigma_{ii}$ for a constant $C$. Given that estimate, we again will call $\text{Avg}_{M,\lambda,\varepsilon,\delta}$, now for a $(d-k)$-dimensional estimation problem. We input the samples $X_{I_{\text{bot}}}^{(1)}, \ldots, X_{I_{\text{bot}}}^{(n)}$, replace the matrix $M$ with the identity of dimension $(d-k) \times (d-k)$, and let $\lambda = O\left(\sqrt{\hat{S}_{\text{bot}} \log \frac{n}{\beta}}\right)$.[6] Denote the output by $\hat{\mu}_{\text{bot}}$. The guarantees of Theorem B.1 provide that with probability $1 - \beta$, $\|\hat{\mu}_{\text{bot}} - \mu_{\text{bot}}\|_2 \leq \alpha$, if, additionally to Eq. (17), we have

$$n \geq \tilde{\Omega}\left(\frac{\sqrt{d \log \frac{1}{\delta} \sum_{i \in I_{\text{bot}}} \Sigma_{ii}}}{\alpha\varepsilon}\right), \tag{18}$$

where $\tilde{\Omega}$ hides multiplicative logarithmic factors of $1/\beta$ and of the term in parentheses. As we prove below, combining these guarantees would yield the desired result. Additionally, we note that in order for the proof to go through, we split the sample into two groups. One group is used for estimating the variances and the other group is given as an input to the two invocations of VarianceSum. We provide the formal pseudocode in Algorithm 2.

**Accuracy analysis.** We put together the statements of the lemmas above, to establish the overall accuracy guarantee of Algorithm 2. By Lemma D.2, the estimates $V_i^{(j)}$ are valid (i.e., at least $4m/5$ of the groups have approximation up to 2 for every coordinate), with probability $1 - \beta$ as long as $m = \Omega(\log \frac{1}{\beta})$. Consequently, Lemma D.4 implies that as long as $m = \Omega(\frac{1}{\varepsilon} \log \frac{1}{\delta\beta})$, FindKthLargestVariance outputs an estimate of the $k$-th largest variance, which is accurate up to a constant factor $C = 8$, with probability $1 - \beta$. By scaling, we can assume that, $\hat{R}/8$, is upper bounded by the $k$-th largest variance. Under this assumption, and as long as $m = \Omega\left(\sqrt{\frac{k \log(1/\delta)}{\varepsilon n}} \log \frac{d}{\beta}\right)$, Lemma D.6 implies that w.p. $1 - \beta$, the output of TopVar, $I_{\text{top}}$, is a set of size $k$, containing indices of elements whose variances are at least $\hat{R}/32$. By Lemma D.5, as long as $m = \Omega\left(\frac{1}{\varepsilon'} \log \frac{1}{\delta'\beta'}\right) = \Omega\left(\frac{\sqrt{k \log(1/\delta)}}{\varepsilon} \log \frac{k}{\delta\beta}\right)$, the estimates $\hat{\Sigma}_{ii}$ to the variances in the indices in $I_{\text{top}}$ are accurate up to a constant factor, with a failure probability of $\beta/k$ for each invocation of this lemma, which sums up to a failure probability of $\beta$. Similarly, the estimate $\hat{S}_{\text{bot}}$ has the same guarantee. If $n$ is large enough to satisfy the requirement of Line 1, then all previous constraints on $m$ are satisfied.

Lastly, the two estimates from $\text{Avg}_{M,\lambda,\varepsilon,\delta}$ suffer an approximation of $\alpha$, each with a failure probability of $\beta$, provided that the conditions on the sample complexity $n$, that are given in Eq. (17) and Eq. (18),

---

[6]We could additionally privately learn the largest variance among $I_{\text{bot}}$, denoted by $\hat{s}$ and set $\lambda = O\left(\sqrt{\hat{S}_{\text{bot}}} + \sqrt{\hat{s} \log \frac{n}{\beta}}\right)$ to decouple $\hat{S}_{\text{bot}}$ from the logarithmic factor, but we choose not to for simplicity, and since we did not optimize for logarithmic factors overall.

hold. By assumption on the sample complexity (Eq. (13),(14)), the guarantee of Eq. (17) indeed holds. It remains to prove that the guarantee of Eq. (18) holds as well. We analyze the term $\sqrt{\sum_{i \in I_{\mathrm{bot}}} \Sigma_{ii}}$. Denote vector $\boldsymbol{\sigma}_{\mathrm{bot}} = (\{\sigma_i\}_{i \in I_{\mathrm{bot}}})$ where $\sigma_i = \Sigma_{ii}^{1/2}$. Then $\sqrt{\sum_{i \in I_{\mathrm{bot}}} \Sigma_{ii}} = \|\boldsymbol{\sigma}_{\mathrm{bot}}\|_2$. By Hölder's inequality,

$$\|\boldsymbol{\sigma}_{\mathrm{bot}}\|_2 \leq \sqrt{\|\boldsymbol{\sigma}_{\mathrm{bot}}\|_1 \|\boldsymbol{\sigma}_{\mathrm{bot}}\|_\infty} \leq \sqrt{\|\boldsymbol{\sigma}\|_1 \|\boldsymbol{\sigma}_{\mathrm{bot}}\|_\infty}.$$

By the guarantees of TopVar and FindKthLargestVariance, except for a failure probability of $O(\beta)$, there exists a universal constant $C > 1$ such that

$$\max_{i \in I_{\mathrm{bot}}} \Sigma_{ii}^{1/2} \leq C\hat{R}^{1/2}.$$

Further, by assumption, $\hat{R}$ is up to a constant the $k$-th largest diagonal element of $\Sigma$, hence,

$$C\hat{R}^{1/2} \leq \frac{1}{k} \sum_{i=1}^{d} \Sigma_{ii}^{1/2} .$$

Substituting this above, we obtain that

$$\sqrt{\sum_{i \in I_{\mathrm{bot}}} \Sigma_{ii}} \leq \frac{1}{\sqrt{k}} \sum_{i=1}^{d} \Sigma_{ii}^{1/2} .$$

Thus, it suffices for the stated sample complexity to additionally satisfy $n = \tilde{\Omega}\left(\frac{\sqrt{d \log(1/\delta)} \sum_{i=1}^{d} \Sigma_{ii}^{1/2}}{\sqrt{k} \alpha \varepsilon}\right)$. Substituting the definition for $k$, we obtain Eq. (15), which completes the proof.

**Privacy analysis.** Notice that the output of the algorithm is obtained by composing multiple differentially private mechanisms. Some of these mechanisms access the estimates $V^{(1)}, \ldots, V^{(m)}$ instead of the original dataset. Yet, since each input datapoint $X^{(i)}$ influences only one vector $V^{(j)}$, this implies that any DP guarantees for algorithms that use the $V^{(j)}$ estimates, directly translate to DP guarantees on the original input dataset.

Notice that the algorithm has $O(1)$ calls to $(\varepsilon, \delta)$-DP mechanisms, and $k$ calls to $(\varepsilon', \delta')$-DP mechanisms: these are the calls to VarianceSum. By Lemma A.6 (advanced composition), the concatenation of all the calls to VarianceSum are together, $(O(\varepsilon), O(\delta))$. By basic composition of the same lemma, composing the resulting composition with the other calls to DP mechanisms, yields an $(O(\varepsilon), O(\delta))$-DP mechanism.

### D.1 The variance estimates are valid: Proof of Lemma D.2

The random variable $(X_i^{(j,2r-1)} - X_i^{(j,2r)})/\sqrt{2\Sigma_{ii}}$ is standard normal. Thus,

$$\sum_{r=1}^{\ell} \frac{(X_i^{(j,2r-1)} - X_i^{(j,2r)})^2}{2\Sigma_{ii}}$$

follows a chi-squared distribution with $\ell$ degrees of freedom. We use the following concentration property of a Chi-squared random variable [45, Lemma 1]: if $Z$ is Chi-squared with $\ell$ degrees of freedom,

$$\Pr[\mathbb{E}[Z]/2 \leq Z \leq 2\,\mathbb{E}[Z]] \geq 1 - 2e^{-c\ell} ,$$

for some constant $c > 0$. Consequently,

$$\Pr\left[\frac{\ell}{2} \leq \sum_{r=1}^{\ell} \frac{(X_i^{(j,2r-1)} - X_i^{(j,2r)})^2}{2\Sigma_{ii}} \leq 2\ell\right] \geq 1 - 2e^{-c\ell},$$

for some constant $c > 0$.

By a union bound over all dimensions $i$, the probability that all dimensions' variance estimates fall within the specified bounds in a single group $j$ is at least $1 - 2d \exp(-c\ell)$. Assuming $\ell \geq C \log d$ ensures this probability is very high (e.g., at least $7/8$ for suitable constant $C$).

Using the Chernoff bound for the binomial distribution, if each group independently satisfies the variance bounds with probability at least $7/8$, then the probability that at least $4/5$ of the groups satisfy the variance bounds is at least $1 - \beta$ for $m = \Omega(\log(1/\beta))$.

## D.2 Finding the indices of the largest variances: Proof of Lemma D.6

We propose an algorithm which, receives estimates $V_i^{(j)}$ for the variances and a threshold $R$, and outputs $k$ indices $i \in [d]$ whose variance is at least $R/C$ for some universal constant $C$. To do so, we use the sparse vector algorithm, which receives a dataset $D$, queries $Q_1(D), \ldots, Q_d(D)$, a threshold $T$ and a natural number $k$. It outputs $k$ indices $i$ such that $Q_i(D) \geq T$ (approximately). In order to use the sparse vector to identify the largest variances, our dataset $D$ will be $V$, the collection of estimates. The query $Q_i(V)$ will capture whether the $i$'th variance is $\Omega(R)$. We define the query

$$Q_i(V) = \frac{1}{m} \left| \left\{ j \colon V_i^{(j)} \geq R/2 \right\} \right| ,$$

and the threshold $T = 1/2$. Intuitively, if $Q_i(V) \geq 1/2$ this means that at least half of the values of $j$, $V_i^{(j)} \geq R/2$, which implies that $\Sigma_{ii} \geq \Omega(R)$, provided that the estimates $V$ are valid. Otherwise, it implies $\Sigma_{ii} \leq R$.

We now formally define the sparse vector algorithm [26, 54, 31], and review its guarantees. See [23, Section 3.6] for a detailed analysis of the sparse vector technique.

---

**Algorithm 3** $\mathrm{Sparse}(D, \{Q_i\}, T, d, \varepsilon, \delta)$, from [23]

---

**Require:** Input is a private database $D$, an adaptively chosen stream of sensitivity $1/n$ queries $Q_1, \ldots$, a threshold $T$, a cutoff point $k$, and privacy parameters $\varepsilon, \delta$.

1: $\hat{T} \leftarrow T + \mathrm{Lap}\left(\frac{2}{\varepsilon n}\right)$
2: $\sigma \leftarrow \sqrt{\frac{32k \ln(1/\delta)}{\varepsilon n}}$
3: $\mathrm{count} \leftarrow 0$
4: $I \leftarrow \emptyset$
5: **for** each query $i$ **do**
6:     $v_i \leftarrow \mathrm{Lap}(\sigma)$
7:     **if** $Q_i(D) + v_i \geq \hat{T}$ **then**
8:         $I \leftarrow I \cup \{i\}$
9:         $\mathrm{count} \leftarrow \mathrm{count} + 1$
10:    **if** $\mathrm{count} \geq k$ **then**
11:        **return** $I$
12: **return** $I$

---

**Lemma D.7** (Sparse guarantees). $\mathrm{Sparse}$ *(Algorithm 3) is $(\varepsilon, \delta)$-differentially private. Let $\beta \in (0, 1)$ and define*

$$\alpha = 2\sigma \left( \log d + \log \frac{2}{\beta} \right) = \sqrt{\frac{128k \ln(1/\delta)}{\varepsilon n}} \left( \log d + \log \frac{2}{\beta} \right).$$

*For any sequence of $d$ queries $Q_1, \ldots, Q_d$ if there are at least $k$ queries $i$ such that $Q_i(D) \geq T + \alpha$, then the following holds with probability $1 - \beta$: the output of Algorithm 3, $I$, is a set of size $k$, and for each $i \in I$, $Q_i(D) \geq T - \alpha$.*

Next, we formally define the algorithm $\mathrm{TopVar}$, to find the indices of the largest variances.

---

**Algorithm 4** $\mathrm{TopVar}_{\varepsilon, \delta}(V, R, k)$

---

**Require:** Variance estimates $V = \{V_i^{(j)}\}_{j \in [m], i \in [d]}$, threshold $R \in \mathbb{R}$, privacy parameters $\varepsilon, \delta \in (0, 1)$, number of indices $k \in \mathbb{N}$.

1: Define queries $Q_i(D)$ for each $i \in [d]$ as:

$$Q_i(D) = \frac{1}{m} \left| \left\{ j \colon V_i^{(j)} \geq R/2 \right\} \right|$$

2: $T \leftarrow 1/2$
3: **return** $\mathrm{Sparse}(V, \{Q_i\}, T, k, \delta)$

---

The privacy guarantees of TopVar follow directly from the guarantees of the sparse vector. Next, we describe how to derive the accuracy guarantees. Notice that if the $V_i^{(j)}$ are valid, then, for any $i$ such that $\Sigma_{ii} \geq R$: for at least $4m/5$ values of $j$, it holds that $V_i^{(j)} \geq R/2$, hence, $Q_i(D) \geq 4/5$. Further, for any $i$ such that $\Sigma_{ii} < R/4$, for at least $4m/5$ values of $j$ it holds that $V_i^{(j)} < R/2$, hence $Q_i(D) \leq 1/5$. Hence, if we set the threshold at $T = 1/2$, and $\alpha = 1/4$, then, for any $i$ output by the algorithm, $\Sigma_{ii} \geq R/4$. Further, if there are at least $k$ indices $i$ such that $Q_{ii} \geq R$, the algorithm will output $k$ indices.

### D.3 Finding the $k$-th largest variance: Proof of Lemma D.4

We propose an algorithm, Algorithm 5, that receives pre-computed variance estimates $V_i^{(j)}$ for each group $j$ and coordinate $i$. The algorithm uses them to compute an estimate for the $k$-th largest variance for each $V^{(j)}$:

$$M_j := \text{k-th largest of } \left\{V_1^{(j)}, \ldots, V_d^{(j)}\right\}_{i \in [d]}.$$

Our algorithm combines all of these estimates in a differentially private manner, using a stable histogram: Algorithm 6. That algorithm splits the real line into buckets, $\{B_b\}_{b \in \mathbb{Z} \cup \{-\infty\}}$. It receives the estimates $M_1, \ldots, M_m \in \mathbb{R}$ and outputs the index $b$ of the bucket that contains the largest number of estimates $M_j$ (approximately).

In our application, we would like to estimate the $k$-th largest variance up to a multiplicative constant factor, hence, we define the buckets as

$$B_b = \begin{cases} [4^b, 4^{b+1}) & b \in \mathbb{Z} \\ \{0\} & b = -\infty. \end{cases}$$

Denote by $b^*$ index of the bucket that contains the $k$-th largest diagonal entry of $\Sigma$. If the estimates $V^{(1)}, \ldots, V^{(m)}$ are valid then, by definition of validity (Definition 4.2), it follows that at least $4m/5$ of the estimates $M_j$ fall into the union $B_{b^*-1} \cup B_{b^*} \cup B_{b^*+1}$. Under this assumption, Algorithm 6 is guaranteed to output one of $b^* - 1$, $b^*$ or $b^* + 1$, with probability $1 - \delta$.

The algorithm for $k$-th largest variance, Algorithm 5, is presented here:

---

**Algorithm 5** FindKthLargestVariance$_{\varepsilon, \delta}(\{V_i^{(j)}\}_{i \in [d], j \in [m]}, k)$

---

**Require:** Pre-computed variance estimates $V_i^{(j)}$ for each group $j$ and each coordinate $i$. Privacy parameters $\varepsilon, \delta > 0$. Integer $k \leq d$. Number of groups $m$.
1: **for** $j \in [m]$ **do**
2: $\quad$ $M_j \leftarrow k$-th largest value among $\{V_1^{(j)}, V_2^{(j)}, \ldots, V_d^{(j)}\}$
3: Define bins $\{B_b\}_{b \in \mathbb{Z} \cup \{-\infty\}}$ by:

$$B_b = \begin{cases} [4^b, 4^{b+1}) & b \in \mathbb{Z} \\ \{0\} & b = -\infty \end{cases}$$

4: $b \leftarrow$ StableHistogram$_{\varepsilon, \delta}(\{M_j\}_{j \in [m]}, \{B_b\})$
5: **return** $\hat{M} = 4^b$

---

We proceed by defining StableHistogram as introduced in [16] and providing its guarantees, and then we conclude with the proof of Lemma D.4. The presentation of StableHistogram is from [13].

---

**Algorithm 6** StableHistogram$_{\varepsilon,\delta}(\{M_i\}, \{B_b\})$, from [16]

---

**Require:** Items $M_1, \ldots, M_m \in \mathcal{U}$. Bins $\{B_b\}_{b \in \mathbb{Z}}$. Privacy parameters $\varepsilon, \delta > 0$.
  1: **for** $b \in \mathbb{Z}$ **do**
  2:     $c_b \leftarrow |\{i : z_i \in B_b\}|$
  3: **for** $b$ with $c_b > 0$ **do**
  4:     $\tilde{c}_b \leftarrow c_b + \mathrm{Lap}(2/\varepsilon)$
  5: $\tau \leftarrow 1 + \frac{2\log(1/\delta)}{\varepsilon}$
  6: Let $b_{\max} = \arg\max_b \tilde{c}_b$, with arbitrary tie breaks
  7: **if** $\tilde{c}_{b_{\max}} \geq \tau$ **then**
  8:     **return** $b_{\max}$
  9: **else**
 10:     **return** $\perp$

---

We use its privacy and accuracy guarantees, proved as Lemma C.1 in [13]:

**Lemma D.8** (Stable Histogram Guarantees). StableHistogram$_{\varepsilon,\delta}$ *(Algorithm 6) is* $(\varepsilon, \delta)$-*differentially private. Suppose that there exists* $b^* \in \mathbb{Z}$ *such that*

$$|\{M_1, \ldots, M_m\} \cap (B_{b^*-1} \cup B_{b^*} \cup B_{b^*+1})| \geq 3m/4 .$$

*There exists a constant* $C > 0$ *such that, for all* $0 < \varepsilon, \beta, \delta < 1$, *if*

$$m \geq \frac{C}{\varepsilon} \log \frac{1}{\delta\beta},$$

*then with probability at least* $1 - \beta$, *the algorithm's output lies in* $\{b - 1, b, b + 1\}$.

The privacy guarantees of Algorithm 5 follow directly from the privacy guarantees of Algorithm 6. For the accuracy guarantees, notice that if the estimates $V^{(j)}$ are valid then at least $4m/5$ of the values $M_j$ fall into the bucket $B_{b^*}$ that contains the true value of the $k$-th largest entry of the diagonal of $\Sigma$. Under this assumption, Algorithm 6 is guaranteed to output, with probability $1 - \beta$, one of $b^* - 1$, $b^*$ or $b^* + 1$. This implies that the output of Algorithm 5 is approximates the target quantity up to a constant, as required.

### D.4 Finding a sum of variances: Proof of Lemma D.5

We propose an algorithm that is similar to Algorithm 5, with a single difference: given each estimate $V^{(j)}$, the algorithm computes

$$M_j = \sum_{i \in I} V_i^{(j)} .$$

The algorithm is summarized below:

---

**Algorithm 7** VarianceSum$_{\varepsilon,\delta}(\{V_i^{(j)}\}_{i \in [d], j \in [m]}, I)$

---

**Require:** Pre-computed variance estimates $V_i^{(j)}$ for each group $j$ and each coordinate $i$. Privacy parameters $\varepsilon, \delta > 0$. Subset $I \subseteq [d]$. Number of groups $m$.
  1: **for** $j \in [m]$ **do**
  2:     $M_j \leftarrow \sum_{i \in I} V_i^{(j)}$
  3: Define bins $\{B_b\}_{b \in \mathbb{Z} \cup \{-\infty\}}$ by:

$$B_b = \begin{cases} [4^b, 4^{b+1}) & b \in \mathbb{Z} \\ \{0\} & b = -\infty \end{cases}$$

  4: $b \leftarrow$ StableHistogram$_{\varepsilon,\delta}(\{M_j\}_{j \in [m]}, \{B_b\})$
  5: **return** $\hat{M} = 4^b$

---

The proof is identical to the proof of Lemma D.4. In order to carry that proof, one has to notice that if $b^*$ is the bucket that contains $\sum_{i \in I} \Sigma_{ii}$ and if the estimates $V^{(j)}$ are valid, then at least $4m/5$ of the estimates $M_j$ fall within $B_{b^*-1} \cup B_{b^*} \cup B_{b^*+1}$.

