# OpenReview forum: "Dimension-free Private Mean Estimation for Anisotropic Distributions"
_NeurIPS.cc/2024/Conference — NeurIPS 2024 poster_

### Official Review · Reviewer_PrhZ · 2024-06-25

**Soundness:** 3
**Presentation:** 3
**Contribution:** 2
**Rating:** 5
**Confidence:** 4

**Summary:**

This paper tackles the problem of DP mean estimation for high-dimensional distributions exhibiting anisotropy, meaning the variances along different directions are highly non-equal. Prior works on this problem were plagued by a "curse of dimensionality", requiring sample complexities at least on the order of the square root of the ambient dimension d, even in cases where the non-private setting permits much lower sample sizes. The authors make two main contributions to address this limitation.

Firstly, for the case when the covariance matrix $\Sigma$ is known, they provide a DP algorithm achieving sample complexity independent of the dimension d. Instead, the sample complexity depends solely on $tr(\Sigma^{1/2})$, which can be substantially smaller than d when the distribution is highly anisotropic. The bound matches the optimal non-private sample complexity up to logarithmic factors. Secondly, the authors develop a DP algorithm for the unknown covariance setting that improves upon prior work by reducing the dimension dependence from $1/2$ to $1/4$, while also depending on properties of the diagonal entries of $\Sigma$.

**Strengths:**

- Paper is clear and well written, though it is mathematically heavy. The analysis, supported by proof sketches, appears technically sound.
- The motivations of improved rates with certain covariance structure are intuitive.

**Weaknesses:**

- The results involving known covariance are straightforward. The benefits comes from (1) application of Tsfadia et al. and (2) rescaled noise adding, which is similar to Aumüller et al.
- There is still a $d^{1/4}$ left.

**Questions:**

Major questions with potential score raising:
- Is it possible to establish a lower bound strictly higher than Theorem 1.2 for the unknown covariance case? This could help identify an intrinsic gap between the known and unknown covariance settings beyond logarithmic factors.
- The authors mention that for special cases of eigenvalue/variance decay, the unknown covariance bound can be improved to nearly match the known case up to log factors. What are the key bottlenecks or assumptions preventing a fully dimension-independent bound for unknown covariance? The remaining $d^{1/4}$ dependence seems to stem from coordinates with smaller variances - is this avoidable under some assumptions on the decay behavior?

Minor questions:
- In line 113, is the last term in the sample complexity bound for unknown covariance correct?

**Limitations:**

Limitations are well addressed.

---

> ### Author Rebuttal · Authors · 2024-08-07
>
> We thank the reviewer for their feedback and interest in our results.
>
> - **Lower bound for the case of unknown covariance**: We are not certain that our results are optimal in the unknown covariance case, even assuming a diagonal covariance matrix, and we agree that identifying an intrinsic gap between the known and unknown covariance would be very interesting (additionally since this gap does not exist for privately learning the mean in the Mahalanobis norm). Known lower bound techniques for private statistical estimation (in particular, the most popular fingerprinting technique for exponential families) does not apply as is to our problem. However, our current intuition is that there is a case for which the trace term is constant and the $d^{¼}$ term is necessary. Therefore, we think that it is possible that the $d^{¼}$ term cannot be improved.
>
> - **When can we achieve a dimension-independent bound under unknown covariance**: Indeed, the additional error in our results for the unknown covariance case stems from the $d-k$ coordinates of smaller variance, where $k$ is the number of top-variance coordinates we choose to learn, and we show that we can always learn up to roughly $\varepsilon^2n^2$ coordinates without increasing the sample complexity. In particular, the error incurred in the mean estimation step is $(\lVert\sigma_{1:k}\rVert_1 + \sqrt{d-k} \lVert \sigma_{k+1:d}\rVert_2) / \varepsilon n$. The first term is already smaller than the optimal $\lVert\sigma_{1:d}\rVert_1$, and in our result we show that the second term can always be upper-bounded by $\sqrt{d}\lVert\sigma_{1:d}\rVert_1/\sqrt{k}$. When this inequality is tight for some $k$, our analysis of this algorithm is tight.
> More generally though, there are interesting cases when the term $\sqrt{d-k}\lVert\sigma_{k+1:d}\rVert_2$ is smaller than the optimal $\lVert\sigma_{1:d}\rVert_1$, for $k$ at most $\varepsilon^2n^2$, and then our results match the ones for known covariance, up to logarithmic factors in $d$. One example is the case of exponentially decaying variances: if $\sigma_i=\sigma_1 e^{i-1}$, then it suffices to learn only the top $k=\log(d)$ variances, because the error incurred by the last term $\sqrt{d-k}\lVert\sigma_{k+1:d}\rVert_2\approx 1/\sqrt{d}$, which is smaller than the optimal $\lVert\sigma_{1:d}\rVert_1=O(1)$. Learning the top $\log(d)$ variances using the sparse-vector technique can be achieved with a polylogarithmic in $d$ number of samples, adding only logarithmic factors to the optimal sample complexity.
> - Line 113 omits several logarithmic factors with respect to $\delta$, $\beta$, but indeed there is a typo in this particular term: the exact dependence is $\log^2(d)+\log^{1.5}(d)/\varepsilon\gamma + \log(d)/\varepsilon\gamma^2$. We will add the full calculation for this case of interest in the appendix.

---

> > ### Comment · Reviewer_PrhZ · 2024-08-09
> >
> > Thanks for the clarification and I have updated my score. I still think a lower bound for unknown covariance (even if not tight) would be convincing.

---

### Official Review · Reviewer_Tkrg · 2024-07-07

**Soundness:** 3
**Presentation:** 3
**Contribution:** 3
**Rating:** 6
**Confidence:** 3

**Summary:**

The paper presents new differentially private algorithms for estimating the mean of high-dimensional data, addressing the inefficiencies of traditional methods that suffer from the "curse of dimensionality." The proposed estimators are tailored for anisotropic subgaussian distributions, where data signals are concentrated in fewer dimensions. These estimators achieve optimal sample complexity that is independent of dimensionality when the covariance is known and improve the sample complexity for unknown covariance cases from $d^{1/2}$ to $d^{1/4}$.

**Strengths:**

The paper studied interesting problems and the writing is clear. The authors give both upper bound and lower bound for the problem.

**Weaknesses:**

1. I understand the space of the main text is limited but there is no conclusion.
2. There is no experimental design to verify their theoretical findings.

**Questions:**

1. The lower bound does not totally match the upper bound. Is it possible to improve the gap?
2. In Theorem 3.1., the range of $\epsilon$ is from 0 to 10. Actually, 10 is a weak privacy guarantee in DP in practice. Why do you take this value?

**Limitations:**

There is no experiment.

---

> ### Author Rebuttal · Authors · 2024-08-07
>
> We thank the reviewer for the positive feedback and suggestions to improve the presentation of our paper.
>
> - We agree that having a conclusion would be preferable and since the final manuscript can be a page longer we will certainly add one (and move discussion for future work from the appendix) in the main body.
> - **Lower bound for the case of unknown covariance**: We are not certain that our results are optimal in the unknown covariance case, even assuming a diagonal covariance matrix, and we agree that this is a very interesting question. We know of special cases (e.g., diagonal matrices whose singular values follow an exponential decay or isotropic matrices) for which our results are optimal up to logarithmic factors. Our intuition is that there is a case for which the trace term is constant and the $d^{¼}$ term is necessary. Therefore, we think that it is possible that the $d^{¼}$ term cannot be improved.
> - **Range of epsilon**: Thank you for pointing this out – we understand that the range $(0,10)$ seems somewhat arbitrary, and we will clarify this in the updated version. Please note that the theorem still holds for $\varepsilon\in(0,1)$, which would correspond to a reasonably strong privacy guarantee. However, we wanted to point out that the theorem holds more generally, for $\varepsilon\in(0,c)$ where $c$ is some constant larger than $1$, since $\varepsilon>1$ is still encountered in real-world applications, and might be a regime of interest for those. The choice of constant is due to approximations we take in the privacy proof and it has not been optimized (so the theorem could possibly hold for $\varepsilon\in(0,c)$, where $c>10$). However, as you mentioned, 10 seems to be large enough already, so we did not find it necessary to explore the exact constant $>10$ for which our proof goes through.
> - **Simulations**: We focus on the theoretical analysis of our algorithms and prove their privacy and accuracy guarantees, but we do expect that the known-covariance algorithm will perform well, since the constants involved in our error are provably small. The results for the unknown covariance case are less conclusive and we agree that some experimentation might be interesting in this case. However, we would like to point out that ours are the first methods that are applicable for $n\leq \sqrt{d}/\varepsilon$, which makes it difficult to choose an appropriate baseline to compare to for smaller sample sizes.

---

### Official Review · Reviewer_HUbD · 2024-07-12

**Soundness:** 4
**Presentation:** 4
**Contribution:** 3
**Rating:** 7
**Confidence:** 4

**Summary:**

This paper considers the problem of DP mean estimation and the focus is on the high dimensional settings where the distribution is nearly low rank (or tr(Sigma)<<d), and the error metric is l_2. Prior work in this setting still requires sqrt(d) samples to achieve any non-trivial error which is sub-optimal. In the known covariances setting this paper achieves sample complexity n = tr(Sigma)/alpha^2 + tr(Sigma^1/2)/(alpha*eps) which can be significantly smaller than sqrt(d) in the high dimension setting. The algorithm filters out the outlier using the FriendlyCore algorithm [62] and leverages the propose-test-release framework. For the unknown covariance, one will need to first have a rough estimate of the covariance such that appropriate noise can be added to guarantee privacy. However dimension free estimation of the covariance is impossible, so the idea is to estimate the top-K coordinate with the largest variances, and simply adding small and isotropic gaussian noise to the remaining directions. Combining the error incurred in the top and bottom coordinate gives a d^(1/4) dependency in the dimensionality. On the lower bound side, they prove their known covariance result is optimal up to logarithmic factors. It is unclear what the optimal sample complexity for the unknown covariance should be.

**Strengths:**

DP mean estimation problem in the high dimensional setting is a fundamental problem in differential privacy, and the paper achieves optimal results for known covariance setting and makes improvement for the unknown covariance meeting. The ideas for unknown covariance seem novel to me. The sample complexity of mean estimation in the unknown covariance setting, posed in this paper, remains an interesting open problem.

**Weaknesses:**

The techniques for the known covariance setting have been developed previously.

**Questions:**

I don't have question for the authors.

---

> ### Author Rebuttal · Authors · 2024-08-07
>
> We thank the reviewer for the positive feedback and interest in our results.

---

### Official Review · Reviewer_XFCk · 2024-07-13

**Soundness:** 3
**Presentation:** 3
**Contribution:** 3
**Rating:** 7
**Confidence:** 1

**Summary:**

This paper presents a new method for estimating the mean of a subgaussian distribution, such that differential privacy is guaranteed (i.e., the final result does not provide too much identifying information about any individual sample).  For the proposed method, in the case of known covariance,  the sample complexity depends only on the trace of the (1/2) power of the covariance matrix. This means that, if the distribution is highly anisotropic (with the variance mostly concentrated in only a few dimensions) then the sample complexity does not depend explicitly on the full dimensionality of the space *at all*, only on the dimensionality of the lower-dimensional space where "most" of the variance lies. This improves on the prior SOTA result, from (Aumüller et al 2023), where the sample complexity explicitly contains a factor of sqrt(d), where d is the full dimensionality of the space. The core algorithm proceeds by first filtering outliers, using a differentially-private filtering mechanism proposed by (Tsfadia et al 2022), and then simply taking the mean of the remaining samples and adding Gaussian noise.

A matching lower bound is also proven, showing that we can't in general do any better asymptotically than O(trace(Σ^(1/2))), up to logarithmic factors.


A variant of the algorithm is proposed for the case of unknown covariance as well, which has sample complexity with an explicit dependence on d^(1/4), which is still an impovement overt the prior SOTA d^(1/2).

**Strengths:**

- The problem being addressed is important, given the increasing relevance of privacy guarantees in machine learning.
- The improvements from prior work are clearly contextualized. Sufficient background information is provided to understand the major points of the paper.
- To my first estimation, the technique appears sound, although this is not my area of expertise and I am not qualified to rigorously evaluate the technical correctness of the proposed algorithm.

**Weaknesses:**

- The order of the presentation could be improved. The preliminaries section could be presented earlier in the paper, before the definitions given are referred to.
- No empirical tests are performed. Including a numerical simulation of the algorithm would increase the reader's confidence in the correctness of the theoretical  results, as well as give an idea of the tightness of the bound in practice.

**Questions:**

- Is it possible to derive a separate, tighter lower bound for the the case of unknown covariance? Or is it possible that we may in the future improve from the d^(1/4) result in this paper?

**Limitations:**

Limitations are addressed adequately in the Future Work section.

---

> ### Author Rebuttal · Authors · 2024-08-07
>
> We thank the reviewer for the positive feedback and suggestions to improve the presentation of our paper.
>
> - Thank you for the suggestion on the presentation! We will consider including definitions from the preliminaries earlier in the introduction to formalize the intuition we want to convey. Please let us know if there was a specific definition that would have been helpful to see earlier.
> - **Lower bound for the case of unknown covariance**: We are not certain that our results are optimal in the unknown covariance case, even assuming a diagonal covariance matrix, and we agree that this is a very interesting question. We know of special cases (e.g., diagonal matrices whose singular values follow an exponential decay or isotropic matrices) for which our results are optimal up to logarithmic factors. Our current intuition is that there is a case for which the trace term is constant and the $d^{¼}$ term is necessary. Therefore, we think that it is possible that the $d^{¼}$ term cannot be improved.
> - **Simulations**: We focus on the theoretical analysis of our algorithms and prove their privacy and accuracy guarantees, but we do expect that the known-covariance algorithm will perform well, since the constants involved in our error are provably small. The results for the unknown covariance case are less conclusive and we agree that some experimentation might be interesting in this case. However, we would like to point out that ours are the first methods that are applicable for $n\leq \sqrt{d}/\varepsilon$, which makes it difficult to choose an appropriate baseline to compare to for smaller sample sizes.

---

> > ### Comment · Reviewer_XFCk · 2024-08-13
> > **Response to Rebuttal**
> >
> > Thank you for responding to my comments.
> >
> > To be more specific about the ordering of the paper, the FriendlyCore procedure; and the distinction between "pure" DP and (\epsilon,\delta)-DP, could be introduced earlier, before being referenced.
> >
> > I am keeping my original "Accept" score.

---

### Official Review · Reviewer_JTxB · 2024-07-15

**Soundness:** 4
**Presentation:** 3
**Contribution:** 3
**Rating:** 7
**Confidence:** 4

**Summary:**

The paper studies mean estimation for multivariate Gaussian distributions. It's well known that this problem under differential privacy suffers from a curse of dimensionality- the sample complexity of estimating the mean (in expected $\ell_2^2$ error) scales with $\sqrt{d}$ where $d$ is the dimension. However, the lower bounds are for isotropic covariance matrices, whereas in the real world, covariance matrices are often far from isotropic. The main intuition is that when covariance matrices are far from isotropic, there is a gap between large and small singular values, and hence the number of 'important dimensions' is relatively small- the object behaves 'more lower-dimensional' than the dimension $d$ indicates.

They consider two settings, the known and unknown covariance cases.

1) In the known covariance case, they show a dimension-independent bound (for approximate DP) that depends instead on the sum of (square roots of) singular values. They argue that this is optimal up to logarithmic factors (and also that no pure DP algorithm can achieve such a dimension-independent bound).

2) For the unknown covariance case, one approach is estimating the covariance and then applying the known-covariance algorithm, but this might be prohibitive since estimating the covariance is known to require at least $d^{3/2}$ samples asymptotically. The authors instead show that it's possible to achieve a bound with a $d^{1/4}$ dependence instead.

For the known covariance case, the authors use the known FriendlyCore algorithm with a carefully specified predicate to remove outliers (points far from the mean in any coordinate), which allows them to bound the sensitivity and then add dimension-independent noise. The lower bounds are adaptations of the packing and fingerprinting approaches used for pure and approximate DP. In the unknown covariance case, they treat large and small singular values differently. Firstly, they learn the identities and values of the top $k$ singular values up to multiplicative factors and use the known covariance algorithm to estimate the mean restricted to these coordinates. For the other coordinates, they use Holder's inequality to bound the $\ell_2$ norm of the singular values. For small singular values, the variance does not need to be as accurately estimated since these directions are less important. Balancing $k$ to optimize the cumulative error gives the sample complexity bound.

**Strengths:**

1) Gaussian mean estimation is a fundamental problem, and this paper suggests a way to deal with the curse of dimensionality that privacy imposes for this problem, by considering non-worst case instances (anisotropic covariances). This is an interesting direction that will likely spawn future work. The paper also does a nice job of suggesting future directions of research in this space.

2) The paper combines known techniques in privacy in clever ways to obtain their upper and lower bounds. The problem of private Gaussian mean estimation has seen lots of prior investigation, and they also do a good job of explaining how a lot of these techniques inherently give dimension-dependent bounds.

**Weaknesses:**

1) While the results are interesting, they mostly follow from applying known techniques from the privacy literature- the additional technical insight of this paper is rather limited.

2) The unknown covariance case as presented was confusing to me- is there an assumption that the covariance matrix is diagonal; this seems to be used in the author's proofs and approaches? This wasn't clearly described in the paper and could use clarification. I don't see how to extend the techniques of the authors to deal with general covariance matrices.

**Questions:**

1) Line 829, I believe the authors mean $\|\| \sigma_{bot} \|\|_2$.

2) Do known lower bounds for estimating the covariance matrix depend on $d$ even with large singular value gaps? Could one hope to get bounds for this problem that depend similarly on the sum of singular values? I am curious if existing results truly rule out the approach of estimating the covariance matrix before using a known covariance algorithm.

**Limitations:**

Yes

---

> ### Author Rebuttal · Authors · 2024-08-07
>
> We thank the reviewer for the positive feedback and interest in our results.
>
> - **The case of unknown general covariance**: Our main result for unknown covariance (Theorem 1.3) includes the sum of the square-roots of the diagonal elements of the covariance matrix, $\sum_{i=1}^d \Sigma_{ii}^{½}$. This way, we present a result for general covariance without making any assumptions on the covariance matrix. However, this sum is more easily interpretable when the covariance is indeed diagonal: the sum is equal to $\mathrm{tr}(\Sigma^{½})$, when the covariance matrix is diagonal, and can be larger otherwise. In the paper, we focus on the case of diagonal covariance in places, for ease of exposition when describing our techniques, to compare with other baseline approaches which only work for diagonal covariance matrices, or to compare with our result for the case of known covariance which includes the $\mathrm{tr}(\Sigma^{½})$ term. We understand how this can be confusing and we will clarify that we make this assumption for simplicity when we do. Thank you for the suggestion.
>
> - Indeed, Line 829 has a typo and this is the correct term. Thank you for noting it!
>
> - **On the approach of privately estimating the covariance matrix before using a known-covariance mean estimation algorithm**: We do not know of any lower bounds specifically aimed at privately estimating a covariance matrix with large singular value gaps. The existing lower bounds for covariance estimation in spectral norm which apply to Gaussian data ([Kamath Mouzakis Singhal 2022], extended to the low-accuracy regime later by [Narayanan 2023]) use almost-isotropic covariance matrices as the lower bound instance, so they do not go through under a large singular value gap assumption. But we would not expect them to: there are cases [Singhal Steinke 2021] where assuming a known large singular value multiplicative gap in the order of O(d^2) allows us to learn the subspace of the top eigenvectors (and subsequently their singular values) with a sample complexity roughly polynomial in the dimension of the subspace, which does not depend on d.
> However, we note that in any case, we expect that going through covariance estimation to achieve our goal would be a lossy step. The reason is that for the case of (almost) identity covariance, the sample complexity of covariance estimation in spectral norm would be $d^{1.5}/\alpha\varepsilon$, whereas our current upper bound for the unknown covariance case is $d/\alpha\varepsilon+d^{0.75}/\alpha^{½}\varepsilon$, which is smaller (and optimal for the regime $\alpha\leq \sqrt{d}$ which is usually of interest). We will update our manuscript to include the discussion.

---

> > ### Comment · Reviewer_JTxB · 2024-08-08
> >
> > Thanks to the authors for their clarifications!

---

### Decision · Program_Chairs · 2024-09-25

**Decision:**

Accept (poster)

**Comment:**

The authors consider the problem of differentially private mean estimation for anisotropic distributions (i.e., assuming the data has high variance only in a few directions or similarly, assuming data has small effective rank). For non-private setting, simple mean gives dimension-independent bound but doing this privately is non trivial. This is an important problem since mean estimation is a subroutine in many private algorithms. The authors consider the known and unknown covariance settings (results for unknown covariance is not tight). Although the paper may not be technically very novel, the results are interesting, non-trivial, and of broad interest in the privacy community. I recommend acceptance.